# Theory and simulations of condensin mediated loop extrusion in DNA

Ryota Takaki[1], Atreya Dey[2], Guang Shi [2] & D. Thirumalai [2✉]

Condensation of hundreds of mega-base-pair-long human chromosomes in a small nuclear volume is a spectacular biological phenomenon. This process is driven by the formation of chromosome loops. The ATP consuming motor, condensin, interacts with chromatin segments to actively extrude loops. Motivated by real-time imaging of loop extrusion (LE), we created an analytically solvable model, predicting the LE velocity and step size distribution as a function of external load. The theory fits the available experimental data quantitatively, and suggests that condensin must undergo a large conformational change, induced by ATP binding, bringing distant parts of the motor to proximity. Simulations using a simple model confirm that the motor transitions between an open and a closed state in order to extrude loops by a scrunching mechanism, similar to that proposed in DNA bubble formation during bacterial transcription. Changes in the orientation of the motor domains are transmitted over ~50 nm, connecting the motor head and the hinge, thus providing an allosteric basis for LE.

[1] Department of Physics, The University of Texas at Austin, Austin 78712, USA. [2] Department of Chemistry, The University of Texas at Austin, Austin 78712, USA. ✉email: dave.thirumalai@gmail.com

How chromosomes are structurally organized in the tight space of the nucleus is a long-standing problem in biology. Remarkably, these information-carrying polymers in humans with more than 100 million base pairs, are densely packed in the 5−10 μm cell nucleus[1,2]. In order to accomplish this herculean feat, nature has evolved a family of structural maintenance of chromosomes (SMC) complexes[3,4] (bacterial SMC, cohesin, and condensin) to enable large scale compaction of chromosomes in both prokaryotic and eukaryotic systems. Compaction is thought to occur by an active generation of a large array of loops, which are envisioned to form by extrusion of the genomic material[5–7] driven by ATP-consuming motors. The SMC complexes have been identified as a major component of the loop extrusion (LE) process[3,4].

Of interest here is condensin, whose motor activity[8], results in active extrusion of loops in an ATP-dependent manner[9]. Let us first describe the architecture of condensin, shown schematically in Fig. 1. Condensin is a ring-shaped dimeric motor, containing a pair of SMC proteins (Smc2 and Smc4). Both Smc2 and Smc4, which have coiled-coil (CC) structures, are connected at the hinge domain. The ATP binding domains are in the motor heads[4,10]. There are kinks roughly in the middle of the CCs[10]. The relative flexibility in the elbow region (located near the kinks) could be the key to the conformational transitions in the CC that are powered by ATP binding and hydrolysis[4,11].

Previous studies using simulations[6,12,13], which were built on the pioneering insights by Nasmyth[5], suggested that multiple condensins translocate along the chromosome extruding loops of increasing length. In this mechanism, the two condensin heads move away from each other extruding loops in a symmetric manner. Cooperative action of many condensins[14] might be necessary to account for the ~(1000−10,000) fold compaction of human chromosomes[15]. The only other theoretical study that predicts LE velocity as a function of an external load[16] is based on a four-state stochastic kinetic model, with minimally twenty

parameters, for the catalytic cycle of the condensin that is coupled to loop extrusion[16]. In sharp contrast, by focusing on the motor activity of condensin through ATP-driven allosteric changes in the enzyme, our theory and simulations support "scrunching" as a plausible mechanism for loop extrusion. Scrunching is reminiscent of the proposal made over a decade ago in the context of the first stage in bacterial transcription that results in bubble formation in promoter DNA[17], which was quantitatively affirmed using molecular simulations[18]. Recently, the scrunching mechanism was proposed to explain loop extrusion[19], which is fully supported by theory and simulations presented here.

We were inspired by the real-time imaging of LE in λ-DNA by a single condensin[9], which functions by extruding loops asymmetrically. To describe the experimental outcomes quantitatively, we created a simple analytically solvable theory, with two parameters, that produces excellent agreement with experiments for the LE velocity as a function of external load. We also quantitatively reproduce the distribution of LE length per cycle measured using magnetic tweezer experiments[20]. The theory and simulations show that for LE to occur there has to be an ATP-powered allosteric transition in condensin involving a large conformational change that brings distant parts (head and the hinge in Fig. 1) of the motor to proximity. We predict that, on an average, the distance between the head and the hinge decreases by conformational ~(22−26) nm per catalytic cycle. These values are in remarkably close to the experimentally inferred values[9]. Simulations using a simple model, with and without DNA, lend support to our findings. Our work strongly suggests that the conformational transitions are driven by a scrunching mechanism in which the motor is relatively stationary, but DNA is reeled in by an allosteric mechanism.

## Results

**Model description.** In order to develop a model applicable to condensin (and cohesin), we assume that condensin is attached to two loci (**A** and **B**) on the DNA (Fig. 1; right panel). Although, we do not explicitly describe the nature of the attachment points our model is based on the idea of scrunching motion, where two distant ends of condensin move closer upon conformational change, triggered by ATP binding. For example, the green and blue sphere may be mapped onto motor heads and hinge, respectively. The structure of condensin-DNA complex in the LE active form is currently unavailable. However, cryo-EM structures for the related cohesin-DNA complex[21] reveal that DNA is tightly gripped by the two heads of cohesin and the subunits (NIPBL and RAD21). When the results of structural studies are integrated with the observation that the hinge domain of the SMC complexes binds to DNA[22–24], we conclude that both condensin and cohesin must use a similar mechanism to engage with DNA. The head domains in these motors interact with the DNA segment that is in proximity whereas DNA binds only transiently to the hinge. We constructed the model in Fig. 1 based in part on these findings.

In state 1, the spatial distance between the condensin attachment points is, $R_1$, and the genomic length between **A** and **B** is $L_1$. Due to the polymeric nature of the DNA, the captured length $L_1$ could exceed $R_1$. However, $R_1$ cannot be greater than the overall dimension of the SMC motor, which is on the order of ~50 nm. Once a segment in the DNA is captured, condensin undergoes a conformational change driven most likely by ATP binding[20], shrinking the distance from $R_1$ to $R_2$ (where $R_2 < R_1$). As a result, the captured genomic length between **A** and **B** reduces to $L_2$ (state 2). Consequently, the loop grows by $L_1 - L_2$. The step size of condensin is $\Delta R = R_1 - R_2$, and extrusion length per step is $\Delta L = L_1 - L_2$. After the extrusion is

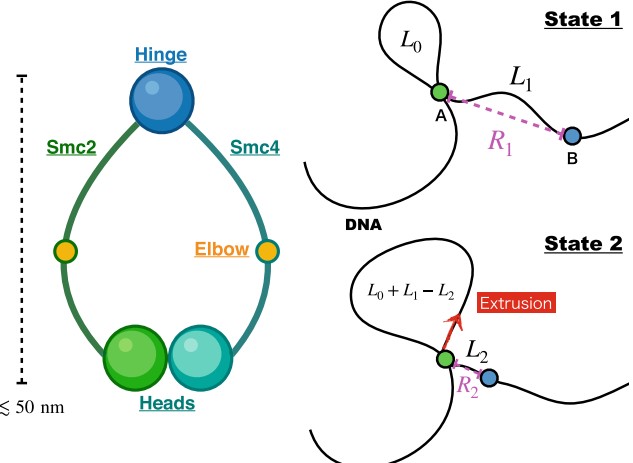

**Fig. 1 Model description.** Left panel: Caricature of the structure of condensin, which has two heads (ATPase domains) and a hinge connected by coiled-coils, labeled Smc2 and Smc4. In the middle of the CCs, there is a flexible kink, referred to as an elbow. Right panel: A schematic of the physical picture for one-sided loop extrusion based on the architecture of a generic SMC complex. DNA is attached to two structural regions on condensin. In state 1 (upper panel) the conformation of condensin is extended with the spatial distance between **A** and **B** equal to $R_1$. The genomic length at the start is $L_0$, which can be large or small. After the conformational transition (state 1 to state 2) the distance between **A** and **B** shrinks to $R_2$, and the length of the extrusion during the single transition is $\Delta L = L_1 - L_2$, which would vary from cycle to cycle.

completed one end of condensin (blue circle in Fig. 1; right panel) is released from the genome segment and starts the DNA capturing process again, likely mediated by diffusion leading to the next LE cycle.

**Theory for the captured length (L) of DNA.** In order to derive an expression for the loop extrusion velocity, we first estimate the loop length of DNA, $L$, captured by condensin when the attachment points are spatially separated by $R$. We show that on the length scale of the size of condensin (~50 nm), it is reasonable to approximate $L \approx R$. To calculate the LE velocity it is necessary to estimate the total work done to extrude DNA with and without an external load. Based on these considerations, we derive an expression for the LE velocity, given by $k_0 \exp(-f\Delta R/k_B T)\Delta R$, where $k_0$ is the rate of mechanical step in the absence of the external load ($f$), $k_B$ is Boltzmann constant, and $T$ is temperature.

*Distribution of captured DNA length without tension, $P(L|R)$.* We examined the possibility that the loop extrusion length per step can be considerably larger than the size of condensin[8–10,25] by calculating, $P(L|R)$, the conditional probability for realizing the contour length $L$ for a given end-to-end distance, $R$. We calculated $P(R|L)$ using a mean-field theory that is an excellent approximation[26] to the exact but complicated expression[27]. The expression for $P(L|R)$, which has the same form as $P(R|L)$, up to a normalization constant, is given by (Sec.III in SI)

$$P(L|R) = A \frac{4\pi N\{L\}(R/L)^2}{L(1-(R/L)^2)^{9/2}} \exp\left(-\frac{3t\{L\}}{4(1-(R/L)^2)}\right), \quad (1)$$

where $t\{L\} = 3L/2l_p$, $l_p$ is the persistence length of the polymer, and $N\{L\} = \frac{4\alpha^{3/2}e^{\alpha}}{\pi^{3/2}(4+12\alpha^{-1}+15\alpha^{-2})}$ with $\alpha\{L\} = 3t/4$. In Eq. (1), $A$ is a normalization constant that does not depend on $L$ where integration range for $L$ is from $R$ to $\infty$. The distribution $P(L|R)$, which scales as $L^{-3/2}$ for large $L$, has a heavy tail and does not have a well defined mean (see Fig. 2a for the plots of $P(L|R)$ for different $R$). The existence of long-tail in the distribution $P(L|R)$ already suggests that condensin, in principle, could capture DNA segment much larger than its size ~50 nm. However, this can only happen with lower probability compared to a more probable scenario where $R \approx L$ near the position of the peak in $P(L|R)$ (see Fig. 2a). Therefore, we evaluated the location of the peak ($L_{peak}$) in $P(L|R)$, and solved the resulting equation numerically. The dependence of $L_{peak}$ on $R$, which is almost linear (Fig. 2a), is well fit using $L_{peak} = R \exp(aR)$ with $a = 0.003 \, \text{nm}^{-1}$ for $R < 60$ nm. Thus, with negligible corrections, we used the approximation $L \approx R$ on the length scales corresponding to the size of condensin or the DNA persistence length. Indeed, the location of the largest probability is at $L \approx l_p \approx R$ ($l_p$ is the persistence length of polymer), which is similar to what was found for proteins[28] as well. The presence of $f$ would stretch the DNA, in turn decrease the length of DNA that condensin captures, further justifying the assumption ($L \approx R$). Therefore, we conclude that $R_1 \approx L_1$ and $R_2 \approx L_2$ (note that $R_2 < R_1 \lesssim l_p^{DNA}$; $l_p^{DNA}$ is the persistence length of DNA), and that LE of DNA loop that is much larger than the size of condensin is less likely. Thus, we expect that the extrusion length of DNA is nearly equal to the step size of condensin, $\Delta R \approx \Delta L$.

*Force-dependent distribution, $P(L, R, f)$, of the captured DNA length.* Following the steps described above, we write the end-to-end distribution of semi-flexible polymer under tension $f$, $P(R, f|L)$, as $P(R, f|L) = BP(R|L)e^{fR/k_B T}$, where $B$ is normalization

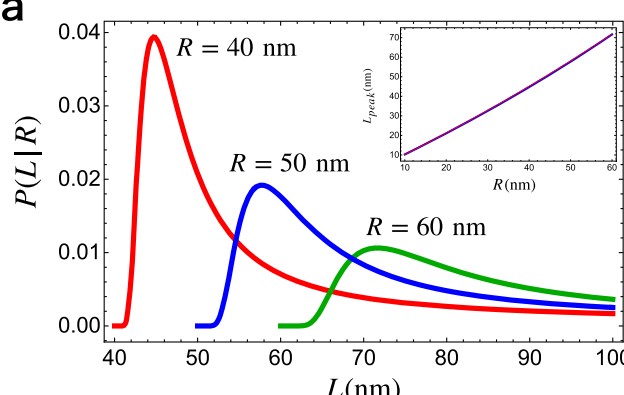

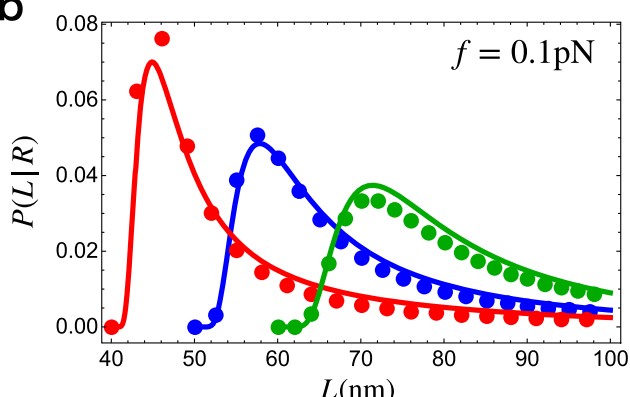

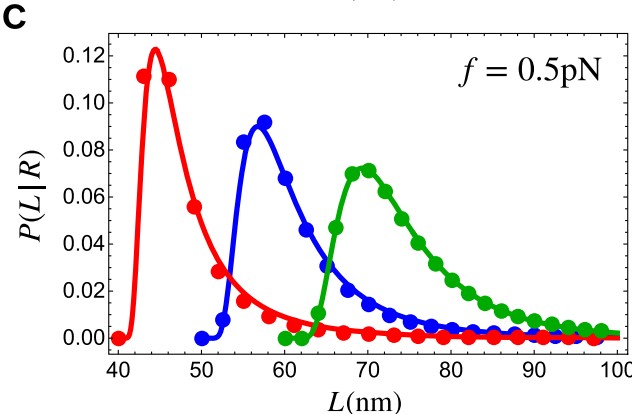

**Fig. 2 Distributions of the captured length of DNA. a** Plots of $P(L|R)$ for different $R$ values; $R = 40$ nm (red), $R = 50$ nm (blue), and $R = 60$ nm (green). Inset: Peak position ($L_{peak}$) of $P(L|R)$, evaluated numerically by setting by d$P(L|R)$/d$L$ to zero, as a function of $R$. The dotted red line is a fit, $L_{peak} = R \exp(aR)$ with $a = 0.003 \, \text{nm}^{-1}$. **b**, **c** The distributions of $L$ for different $R$ and $f$. $R = 40$ nm (red), $R = 50$ nm (blue), and $R = 60$ nm (green). The dots are from Eq. (2) and the solid lines are the approximate probability distribution Eq. (3). We used the persistence length of DNA, $l_p^{DNA} = 50$ nm.

constant. Thus, $P(L|R, f)$ is obtained using,

$$P(L|R, f) = CB\{L\}\frac{4\pi N\{L\}(R/L)^2}{L(1-(R/L)^2)^{9/2}}\exp\left(-\frac{3t\{L\}}{4(1-(R/L)^2)}\right)\exp\left(\frac{fR}{k_B T}\right), \quad (2)$$

where $C$ is a normalization constant that does not depend on $L$. The constant $B\{L\}$ for $P(R, f|L)$, which carries the $L$ dependence, prevents us from deriving an analytically tractable expression for $P(L|R, f)$. We find that, for a sufficiently stiff polymer, ($R/l_p \lesssim 1$),

$P(L|R, f)$ is well approximated by,

$$P_A(L|R, f > 0) = D \frac{4\pi N^2\{L\}(R/L)^2}{L(1-(R/L)^2)^{9/2}} \exp\left(-\frac{3t\{L\}}{4(1-(R/L)^2)}\right)$$
$$\times \exp\left(\frac{fR}{k_B T} - \left(1.0 + 3.3e^{-f/f_0}\right)\frac{fL}{k_B T}\right),$$
(3)

where $f_0 = 1/7$ pN, $t\{L\} = 3L/2l_p$, $l_p$ is the persistence length of the polymer, and $N^2\{L\} = \left(\frac{4\alpha^{3/2}e^\alpha}{\pi^{3/2}(4+12\alpha^{-1}+15\alpha^{-2})}\right)^2$ with $\alpha\{L\} = 3t/4$. The constant $D$ does not depend on $L$ in the integration range. Note that on the scale of condensin DNA is relatively rigid ($R \lesssim l_p^{DNA} \sim 50$ nm). Probability Eqs. 2 and 3 for different values of $f$ in Fig. 2 show that the agreement between $P_A(L|R)$ and $P(L|R, f)$ is good. Therefore, in what follows we use Eq. (3). The distributions for $f > 0$ in Fig. 2 show that the position of the peak for $P(L|R, f)$ do not change over the range of $f$ of interest. However, the height of the peak increases as $f$ increases accompanied by the shrinking of the tail for large $L$.

**Condensin converts chemical energy into mechanical work for LE.** Just like other motors, condensin hydrolyzes ATP, generating $\mu \approx 20$ $k_B T$ chemical energy that is converted into mechanical work, which in this case results in extrusion of a DNA loop[9]. To derive an expression for LE velocity, we calculated the thermodynamic work required for LE. The required work $W$ modulates the rate of the mechanical process by the exponential factor $\exp(-W/k_B T)$. In our model, $W$ has two contributions. The first is the work needed to extrude the DNA at $f = 0$ ($W_0$). Condensin extrudes the loop by decreasing the spatial distance between the attachment points from $R = R_1$ to $R = R_2$ (Fig. 1). The associated genomic length of DNA that has to be deformed is $L_\Sigma = L_0 + L_1$. The second contribution is $W_f$, which comes by applying an external load. Condensin resists $f$ up to a threshold value[9], which may be thought of as the stall force. The mechanical work done when condensin takes a step, $\Delta R = R_1 - R_2$, is $W_f = f\Delta R$.

We calculated $W$ as the free energy change needed to bring a semi-flexible polymer with contour length $L_\Sigma$, from the end-to-end distance $R_1$ to $R_2$. It can be estimated using the relation, $W \approx -k_B T \log\left(P(R_2, f|L_\Sigma)\right) + k_B T \log\left(P(R_1, f|L_\Sigma)\right)$, where $P(R, f|L)$ is given by $P(R, f|L) = BP(R|L)e^{fR/k_B T}$ where $B$ is a normalization constant. Although $P(R, f|L)$ is a distribution, implying that there is a distribution for $W$, for illustrative purposes, we plot $W$ in Fig. 3 for a fixed $R_1 = 40$ nm and $R_2 = 14$ nm corresponding to $\Delta R = 26$ nm as estimated using our theory to experiment in later section. It is evident that condensin has to overcome the highest bending penalty in the first step of extrusion, and subsequently $W$ is essentially a constant at large $L_\Sigma$. Note that when $f = 0$ (red line in Fig. 3), $W = W_0$ because the $W_f$ term vanishes. If $R_1 = 40$ nm, which is approximately the size of the condensin motor, we estimate that condensin pays $5k_B T$ to initiate the extrusion process without tension (red line in Fig. 3).

Once the energetic costs for LE are known, we can calculate the LE velocity as a function of an external load applied to condensin. From energy conservation, we obtain the equality, $n\mu = W_0 + W_f + Q$, where $n$ is the number of ATP molecules consumed per mechanical step, $\mu$ is the energy released by ATP hydrolysis, and $Q$ is the heat dissipated during the extrusion process. The maximum force is obtained at equilibrium when the equality $n\mu = W_0 + W_f\{f_{max}\}$ holds. If we denote the rate of mechanical transition as $k^+$ and reverse rate as $k^-$, fluctuation theorem[29–31] together with conservation of energy gives the following relation:

$$k^+/k^- = e^{(n\mu - W_0 - W_f)/k_B T}.$$
(4)

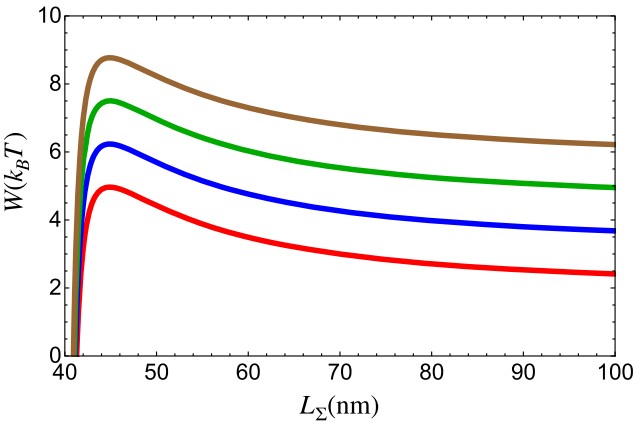

**Fig. 3 Energetic cost to extrude DNA.** $W = W_0 + W_f$, as a function of $L_\Sigma$ (~total extruded length of DNA; $L_\Sigma = L_0 + L_1$) for different $f$ values on DNA. We set $R_1 = 40$ nm as the size of condensin in open state (State 1 in Fig. 1) and $R_2 = 14$ nm, which corresponds to $\Delta R = 26$ nm as obtained from the fit of our theory to experiment[9] in later section. $f = 0$ pN is in red, $f = 0.2$ pN is in blue, $f = 0.4$ pN is in green, and $f = 0.6$ pN is in brown. We used the persistence length of DNA, $l_p^{DNA} = 50$ nm.

Note that condensin operates in non-equilibrium by transitioning from state 1 to 2, which requires input of energy. The load dependent form for the above equation may be written as,

$$k^+ = k_0 e^{-W_f/k_B T},$$
(5)

where $k_0 = k^- e^{(n\mu - W_0)/k_B T}$ is the rate of the mechanical transition at 0 load. Thus, with $\Delta R$ being the extruded length per reaction cycle, the velocity of LE, $\Omega$, may be written as,

$$\Omega\{f\} = k_0 e^{-f\Delta R/k_B T}\Delta R.$$
(6)

It is worth stating that $k_0$ is the chemical energy dependent term, which includes $\mu$, depends on the nucleotide concentration. In order to obtain the ATP dependence, we assume the Michaelis–Menten form for $k_0$, $\frac{\hat{k}_0[T]}{[T]+K_M}$, where $[T]$ is the concentration of ATP, $\hat{k}_0$ is the maximum rate at saturating ATP concentration, and $K_M$ is the Michaelis–Menten constant (see Fig. 4b). In principle, $K_M$ should be determined from the measurement of LE velocity as a function of $[T]$. Here, we use $K_M = 0.4$mM obtained as Michaelis–Menten constant for ATP hydrolysis rate[8], assuming that ATP hydrolysis rate is the rate-limiting step in the loop extrusion process.

In order to calculate $\Omega$ as a function of the relative DNA extension, $x$, we use the expression[32,33],

$$f = \frac{k_B T}{2l_p}\left[2x + \frac{1}{2}\left(\frac{1}{1-x}\right)^2 - \frac{1}{2}\right].$$
(7)

The dimensionless variable, $x$, is the $f$-dependent relative extension. In the Ganji et al.[9] experiment $x$ is measured, and the $f$-dependence of $\Omega$ is obtained by expressing $x$ in terms of $f$ using a numerical procedure.

**Analysis of experimental data**

*Loop extrusion velocity*. We used Eq. (6) to fit the experimentally measured LE velocity as a function of DNA extension[9]. The two fitting parameters are $\Delta R$, and $k_0$, the average step size for condensin, and the extrusion rate at $f = 0$, respectively. Excellent fit of theory to experiments, especially considering the dispersion in the data, gives $k_0 = 20$ s$^{-1}$ and $\Delta R = 26$ nm. This indicates that condensin undergoes a conformational change that brings the head and the hinge to within $\Delta R \sim 26$ nm (~76 bps), during each extrusion

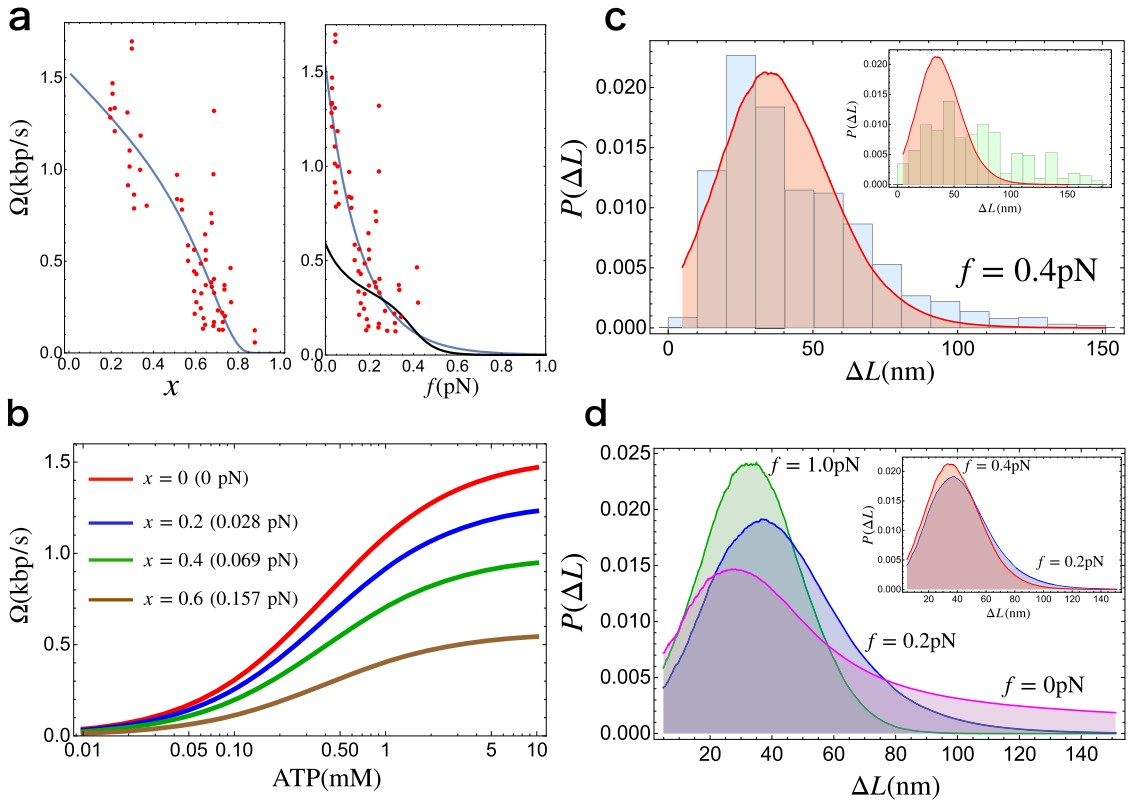

**Fig. 4 Analysis of experimental data using theory. a** Left panel: The LE velocity as a function of the relative extension of the DNA ($x$). Red dots are experimental data[9], and the solid blue line is the fit obtained using Eq. (6). We used $l_p^{DNA} = 50$ nm for the persistence length of DNA. Right panel: Extrusion velocity as a function of the external load acting on DNA ($f$). The blue line is $\Omega$ from Eq. (6) and the line in black is reproduced from Fig. 6c in ref. [16]. The unit of LE velocity, nm/s, is converted to kbp/s using the conversion 1 bp = 0.34 nm. **b** The dependence of LE velocity on ATP concentration for different relative extension of DNA. The parameters used are $\hat{k}_O = 20$ s$^{-1}$, $\Delta R = 26$ nm, and $K_M = 0.4$ mM, where $K_M = 0.4$ mM is Michaelis–Menten constant for ATP hydrolysis rate[8]. We used $l_p^{DNA} = 50$ nm for the persistence length of DNA. **c** Distribution of the DNA extrusion length per step ($\Delta L$) using $l_p^{DNA} = 42$ nm, which is the value reported in the experiment[20]. The histograms are the experimental data taken from Ryu et al.[20] (blue) and Strick et al.[58] (inset;green). The distributions in red are the theoretical calculations. The distributions for theory and the experiments are both in $f = 0.4$ pN. **d** LE length distributions for various external load on DNA using $l_p^{DNA} = 42$ nm[20]. $f = 0$ pN is in magenta, $f = 0.2$ pN is in blue, and $f = 1.0$ pN is in green. The inset compares the results for $f = 0.2$ pN and $f = 0.4$ pN. Data plotted in **a**, **c**, and **d** are provided as a Source Data file.

cycle. This prediction is remarkably close to the value measured in the recent AFM experiment ~22 nm[19], and is further supported by our simulations (see below). We note that $k_0 = 20$ s$^{-1}$ is roughly ten times greater than the bulk hydrolysis rate estimated from ensemble experiments[8,9]. A plausible reason for the apparent discrepancy, already provided in the experimental studies[8,9], is that bulk hydrolysis rate could underestimate the true hydrolysis rate due to the presence of inactive condensins. Thus, the estimated rate of $k_0 = 2$ s$^{-1}$ should be viewed as a lower bound[9]. Another possible reason for the discrepancy may be due to methods used to estimate $k_0$ in previous experiments[8,9]. It is clear that additional experiments are needed to obtain better estimates of the hydrolysis rate, which is almost all theories is seldom estimated. In Fig. 4a right panel we compare the dependence of $\Omega$ on $f$ obtained in a previous kinetic model that has in excess of 20 parameters[16]. In contrast to our theory, even the shape of the LE velocity for $\Omega$ versus $f$ does not agree with experiment. In addition, there is a major discrepancy (factor of 2–3) between the predicted and the measured values of $\Omega$ at low force.

*LE length distribution.* Recently Ryu et al.[20] measured the distribution of LE length per step using magnetic tweezers. We calculated the LE distribution using Eq. (1) and Eq. (3). Of interest here is the distribution for $L_1 - L_2$ where $L_1$ is the captured length of DNA in open shape (see Fig. 1; right panel). We

use the length of condensin in the open state as $R_1 = 40$ nm (roughly the peak in the distance between the head and the hinge in the O shape in the wild type condensin), and assume that $\Delta R = 26$ nm during a single catalytic cycle, as theoretically calculated in the previous section. This gives the length of the closed state, $R_2 = 14$ nm. Previous experiments[19] reported significant fluctuations in the size of the open and closed states, leading to the standard deviation for the change between these two states to be $\Delta = 13$ nm. For simplicity, we include the standard deviation for the conformational change in the open state $R_1 \pm \Delta$ and keep $R_2$ fixed. This is justified because as $R$ decreases ($R_2 < R_1$), not only does the peak in $P(L|R)$ and $P_A(L|R,f)$ moves to small $L$ but also there is a decrease in the fluctuation (width of the distribution is smaller) (see Fig. 2). This suggests that the variance of the extrusion length owing to $R_2$ is negligible. We assume that the distribution of $R_1$ is a Gaussian centered at 40 nm with a standard deviation of 13 nm. It is reasonable to approximate $L_2 \approx R_2$ since $R_2 = 14$ nm $< l_p^{DNA} = 50$ nm. With these parameters in hand, the captured DNA length, Eq. 1 and 3, directly lead to an expression for the distributions for LE length per step. The probability for the LE length per step ($\Delta L$) is, $P(\Delta L) = P(L + R_2|R_1)$ for $f = 0$ and $P(\Delta L|f) = P_A(L + R_2|R_1,f)$ for $f > 0$.

We compare in Fig. 4 the theoretically calculated distribution for $f = 0$ pN from Eq. (1) and $f = 0.4$ pN from Eq. (3) with experiments. The distribution for $f = 0$ pN cannot be measured

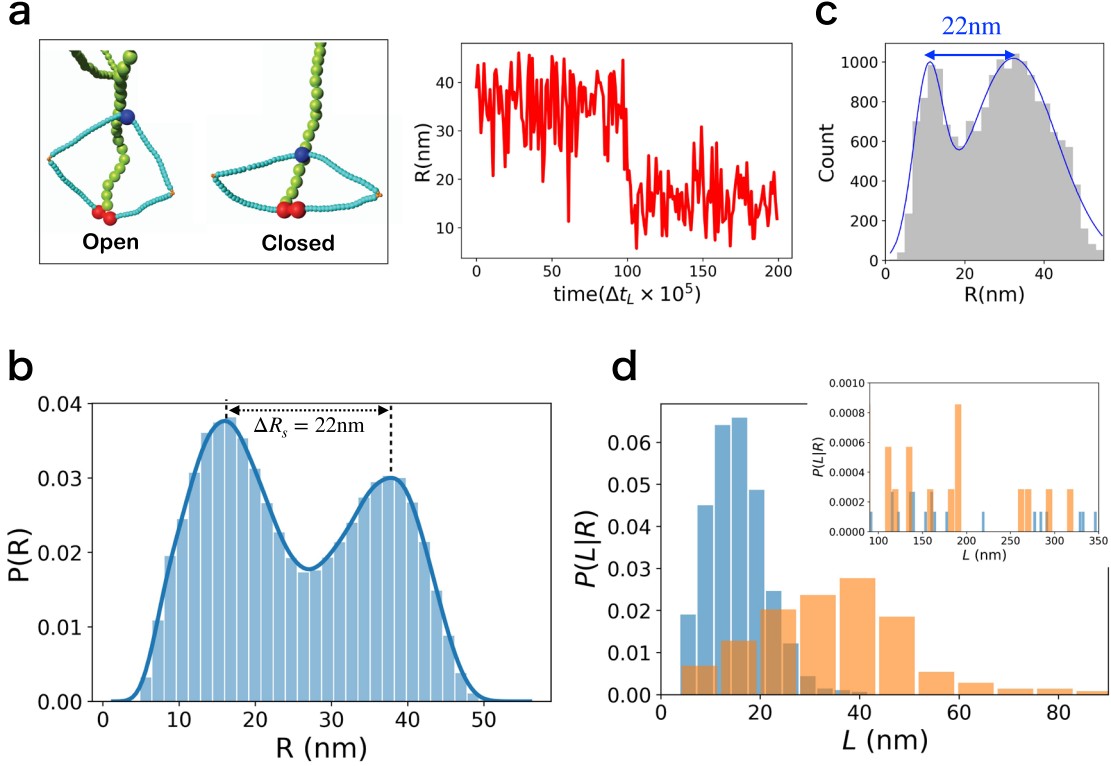

**Fig. 5 Simulations for the transition between O → B transition. a** Left panel: Representative pictures from the simulation. Red, blue, light blue, orange, and green spheres are the heads, hinge, CC, elbows, and DNA, respectively. Right panel: The trajectory for the change in head-hinge distance ($R$). $\Delta t_L$ is the time step of simulation (SI Sec. I). **b** Predicted distributions, $P(R)$s, of the head-hinge distance during one catalytic cycle of the motor. The peak positions for open state and closed state ($R_1 = 38$ nm and $R_2 = 16$ nm, respectively). The large width and the overlap between the two distributions implies a great deal of conformational heterogeneity that contributes also to broad step size distribution. **c** The distribution for $R$ taken (digitized) from Ryu et al.[19]. The agreement between the simulations and experiments is remarkable, especially considering that the model has no fitting parameters. **d** Histograms in orange are the distribution of $L$ in the open shape, and the blue histograms are the distribution in the closed shape. The inset shows the distributions for $L > 90$ nm. We performed 50 simulations from which 100,000 sample points were used to create the histograms. Data plotted in **a–d** are provided as a Source Data file.

using magnetic tweezers but provides insights into the range of LE length that condensin could take at $f = 0$. Remarkably, the calculated distribution is in excellent agreement with the experimental data[20].

**Plausible conformational change of condensin in LE process**
*Simulations without DNA.* Next we tested whether the predicted value of $\Delta R \sim 26$ nm, which is in fair agreement with the experiment, is reasonable using simulations of a simple model. Because the ATPase domains are located at the heads of condensin, it is natural to assume that the head domain undergoes conformational transitions upon ATP binding and/or hydrolysis. Images of the CCs of the yeast condensin (Smc2–Smc4) using liquid atomic force microscopy (AFM) show they could adopt a few distinct shapes[19,34]. Based on these experiments, we hypothesize that the conformational changes initiated at the head domain result in changes in the angle at the junction connecting the motor head to the CC that propagates through the whole condensin via the CC by an allosteric mechanism[35]. The open (O-shaped in ref. [19]), with the hinge that is ≈40 nm away from the motor domain, and the closed (B-shape[19]) in which the hinge domain is in proximity to the motor domain are the two relevant allosteric states for LE[19,34]. To capture the reaction cycle (O → B → O), we model the CCs as kinked semi-flexible polymers (two moderately stiff segments connected by a flexible elbow), generalizing a similar description of stepping of Myosin V on actin[36]. By altering the angle between the two heads the allosteric

transition between the open (O-shaped) and closed (B-shaped) states could be simulated (SI contains the details).

We tracked the head-hinge distance ($R$) changes during the transition from the open ($R = R_1$) to the closed state ($R = R_2$) in order to calculate the distribution of $\Delta R_s = R_1 - R_2$. The sample trajectory in Fig. 5a, monitoring the conformational transition between the two states, shows that $\Delta R_s$ changes by ~22 nm for the persistence length of condensin ($l_p^{CC}$; see Sec. II in SI for the detail) $l_p^{CC} = 24$ nm, which roughly coincides with the value extracted by fitting the theory to the experimental data. Higher (smaller) values of $\Delta R_s$ may be obtained using larger (smaller) values of $l_p^{CC}$ (Supplementary Fig. 3 in SI). The distributions, $P(R)$, calculated from multiple trajectories (Fig. 5b) are broad, suggestive of high degree of conformational heterogeneity in the structural transition between the open and closed states. The large dispersions found in the simulations is in surprisingly excellent agreement with experiments[19], which report that the distance between the peaks is $22 \pm 13$ nm. We find that the corresponding value is $22 \pm 9$ nm. The uncertainty is calculated using standard deviation in the distributions. Overall the simulations not only provide insight into the physical basis of the theory but also lend support to recent single molecule experiments[19].

*Simulations with DNA.* The purpose of the simulations discussed in the previous section was to assess whether the allosteric mechanism produces a structural rationale for the value of

$\Delta R \sim 26$ nm extracted from the theory. We also created a simple model of condensin with DNA to give additional insights into the DNA-capture mechanism, which is directly related to the extrusion length of DNA per step. We assume that the capture length of DNA by condensin, $L$, is governed by diffusion of the hinge domain, and that the $L$ is solely determined by the semi-flexible polymer nature of DNA. We attached one end of the DNA to the heads of condensin. The other end of DNA diffuses freely during the simulations. We define the DNA "capture" event by the distance between a DNA segment and the condensin hinge, with a cut-off length of 4 nm: if the distance is less than 4 nm we assume that condensin captures the DNA segment. Captured DNA length is the contour length of DNA held between the heads and the hinge. We used a coarse-grained bead-spring model for DNA[26,37]. Each bead represents ten base-pairs, which implies that the bead size is $\sigma_{DNA} = 3.4$ nm. The chain has $N = 100$ beads or 1000 base-pairs. The simulation model for DNA is in the SI (Sec. I). The simulations with DNA explain an interesting aspect of the DNA capture process. In contrast to well-studied molecular motors that take a step that is nearly constant (conventional kinesin and myosin V) and walks on rigid linear track (microtubule and actin filament, respectively), condensin could in principle capture variable length of DNA during each catalytic cycle because it is a flexible polymer unlike microtubule. Figure 5d shows that there is a finite probability that $L$ exceeds the position of the peak by a considerable amount, as predicted in Fig. 2a. This implies that the $L_1$ can be as large as $(60-100)$ nm $(\sim(180-290)$bps$)$, which allows for condensin to extrude substantial length of DNA in each catalytic cycle (see the distributions $P(\Delta L)$ in Fig. 4).

Our results show that theory and simulations for the LE velocity [Eq. (6)] predicts the extent of the conformational change of condensin during the LE process fairly accurately ($\sim 26$ nm in theory and $\sim 22$ nm in the experiment[19] and simulations) and gives the distributions of LE length per cycle in good agreement with experiment. Although our theory is in good agreement with experiments for load-dependent LE velocity and distribution of step sizes, the calculated persistence length (24 nm) of the SMC coils is much higher than the value ($\sim 3.8$ nm) estimated from analyses of AFM images in combination with simulations using the worm-like chain model[34]. It is possible that due to interaction between condensin and other proteins (Brn1 for example) could constrain the head movement, and thus stiffen the coiled-coil. However, the discrepancy between theoretical predictions and simulations is too large to be explained by such effects. Despite performing many simulations using a variety of polymer models, the origin of this discrepancy is unclear. Simulations with flexible coiled-coil do not reproduce the measurements of quantities such as $\Delta R$, that are directly monitored in single molecule experiments[19] (see Fig. S3). We believe that additional experiments and simulations based on higher resolution structures in different nucleotide states are required to close the gap.

## Discussion

**Connection to experiments**. Two of the most insightful experimental studies[19,20] have reported the mean LE velocity and step size distribution as a function of $f$. Because these are direct single molecule measurements that have caught the motor in the act of LE, the results are unambiguous, requiring little or no interpretation. Minimally any viable theory must account for these measurements as quantitatively as possible, using only a small number of physically meaningful parameters. To our knowledge, our theory is currently the only one that reproduces the experimental observations accurately with just two parameters. The only other theory[16] that reported LE velocity as a function of

force, not only has a large number of parameters but also it cannot be used to calculate the step size distribution.

We first showed that the calculated LE velocity could be fit to experimental data in order to extract the hydrolysis rate and the mean step size. To provide a physical interpretation of the theoretically predicted mean step size, we performed polymer based simulations using a simple model of the CCs. The model for the simulations was based on the AFM images[19], which showed that, during loop extrusion, there is a transition between the O shape (head and the hinge are far apart) to the B shape where they are closer. Remarkably, our simulations capture the distributions of the head-hinge distances in the O and B states well without any fitting parameters. The mean distance between the peaks ($\Delta R_s \approx 22$ nm) in the simulations is in excellent agreement with measured value (see Fig. 2c in ref. [19]). It is worth emphasizing that our theoretical fit to experiment yielded $\Delta R \sim 26$ nm, which is also in very good agreement with $\sim 22$ nm measured in the high speed AFM imaging experiment[19]. Thus, both experiments and simulations support the mechanism that repeated O $\rightarrow$ B shape transitions result in extrusion of the DNA loops.

**Relation to a previous study**. Recently, a four state chemical kinetic model[16], similar to the ones used to interpret experiments on stepping of myosin and kinesin motors on polar tracks (actin and microtubule)[31,38], was introduced in order to calculate the $f$-dependent LE velocity and the loop size. The agreement between the predicted dependence on LE velocity and loop size as a function of $f$ is not satisfactory (see Fig. 4a). Apart from the very large number of parameters (about twenty one in the simplified version of the theory[16]), our two parameter theory differs from the previous study in other important ways. (1) The model[16] is apparently based on the rod-like (or I shape) X-ray structure of the prokaryotic CC of the SMC dimer[10], which was pieced together by joining several segments of the CC. However, the theory itself does not incorporate any structural information but is based on a number of rates connecting the four assumed states in the reaction cycle of the motor, and energetics associated with the isolated DNA. (2) Because the previous purely kinetic model[16] does not explicitly consider the structure of condensin, it implies that an allosteric communication between the hinge and the head—an integral part of our theory and observed in AFM experiments[34], is not even considered for the LE mechanism[16]. The lack of conformational changes in response to ATP-binding implies that the substantial decrease in the head-hinge distance by about $\sim 22$ nm observed in AFM imaging experiments cannot be explained, as was noted previously[19]. (3) In the picture underlying the DNA capture model[16] (referred to as the DNA pumping model elsewhere[19]), the distance between the head and the hinge changes very little, if at all. Such a scenario is explicitly ruled out in an experimental study by Ryu et. al.[19] in part because they seldom observe the I shape in the holocomplex by itself or in association with DNA. For this reason, we believe that the mechanism proposed in the recent simulation study[39] is unlikely to be viable. Rather, it is the O $\leftrightarrow$ B transition that drives the loop extrusion process, as found in experiments[19], and affirmed here using our simulations.

**Structural basis for LE**. The paucity of structures for condensin and cohesin in distinct nucleotide bound states makes it difficult to interpret experiments, theory and simulations in molecular terms. The situation is further exacerbated because even the biochemical reaction cycle of condensin (or the related motor cohesin) has not been determined. A recent 8.1 Å structure of the yeast condensin holocomplex in both the *apo* non-engaged state

(the one in which the regulatory element, YCS4, bring the heads in proximity), and the *apo*-bridged state in which the heads interact with each other show a sharp turn in the elbow region, resembling an inverted letter J in the representation in Fig. 1 in a recent study[40]. The functional importance of the inverted J state is unclear because when condensin is active (extruding loops in an ATP-dependent manner) only the O-shaped and B-shaped structures are apparently observed[19]. Furthermore, the structure of the related motor, cohesin, with DNA shows that the heads are bound to DNA with the hinge is in proximity[21], which is inconceivable if the CCs adopt only the I shape. For this reason, we compared our simulations directly with experiments that have measured the distance changes between the head and the hinge during the active LE process[19].

The partially resolved structure of the ATP-bound state shows a large opening of the CC near the heads[40], suggestive of an allosterically driven conformational change. In contrast, the structure of only the prokaryotic SMC coiled-coil at 3.2 Å resolution, which was created by piecing together several fragments in the CC, showed that it adopts the I shape. As noted elsewhere[19], the I-shaped structure is almost never observed in yeast condensin during its function. The paucity of structures prevents any meaningful inclusion of structural details in theory and simulations. It is for this reason, we resorted to comparisons to AFM imaging data and results from single molecule magnetic tweezer experiments, which have caught the yeast condensin as it executes its function, in validating our theory. After all it is the function that matters. In the SI (Sec. VI), we performed structural alignment and normal mode analysis using the partially available cryo-EM structures[40] in order to capture the possible conformational transition in the *apo* state that is poised to transition to the LE active state. Even using only partially resolved structures, the normal mode analyses show that ATP binding induces a substantial opening in the CC region that interacts with the head domains. This preliminary analysis does suggest that for loop extrusion to occur there has to be an allosteric mechanism that brings the head and hinge of the motor close to each other spatially.

**Scrunching versus translocation**. Using a combination of simulations based on a simple model and theory, we have proposed that LE occurs by a scrunching mechanism. The crux of the scrunching mechanism is that the motor heads, once bound to the DNA, are relatively stationary. Extrusion of the loop occurs by the change in the distance between the head and the hinge by about ~22 nm. This conformational change is likely driven by ATP binding to condensin[20]. As a result, the head reels in the DNA, with the mean length that could be as large as ~100 nm, although the most probable value is $\approx(25-40)$ nm depending on the external load (Fig. 4c, d). Recent experiments have suggested that LE occurs by a scrunching mechanism[20], although it was (as stated earlier) proposed in the context of DNA bubble formation[17], which is the initial stage in bacterial transcription. The near quantitative agreement with experiments for load-dependent LE velocity, and step size distribution shows that the theory and the mechanism are self-consistent.

In contrast, the other mechanism is based on the picture that condensin must translocate along the DNA in order to extrude loops[15]. Elsewhere[9] it is argued that the translocation observed in the experiment is an artifact due to salt and buffer conditions, thus casting doubt on the motor translocating along DNA.

**Directionality of LE**. Directionality is a vital characteristic of biological molecular machines with SMC proteins being no exception. In our model (Fig. 1), the unidirectional LE arises

during two stages. One is the DNA capture process and the other is the actual active loop extrusion (State 1 to State 2 in Fig. 1). Directional loop extrusion emerges as a result of binding of the SMCs and the associated subunits to DNA via anisotropic interactions. Indeed, the structure of cohesin[21] suggests that the subunit (STAG1) interacts with the hinge of cohesin by interactions that are anisotropic. This implies that the directional LE could be set by the very act of binding of the condensin motor heads to DNA, which poises the hinge to preferentially interact with DNA downstream of the motor head, we show schematically as point A in Fig. 1. This in turn ensures that the probability of capture of DNA resulting in $\Delta L < 0$ is minimized.

In addition, in our theory there is an asymmetry in the expressions for $k^+$ and $k^-$ in Eq. (4) arising from the free energy, $\mu$, due to the ATP hydrolysis. This process leads to a decrease in the head-hinge distance, as seem in the liquid AFM images of condensin in action. However, we believe that, with small probability, $\Delta L < 0$ could arise because of the slippage of DNA from the extruded DNA by the strong resistive force on the motor[20]. This situation is reminiscent of the slippage of kinesin on microtubule under resistive load[41]. In the absence of any opposing force, the probability that $\Delta L < 0$ is likely to be small.

**Brownian ratchet and power-stroke**. Whether SMC proteins employ Brownian ratchet mechanism[42] or power stroke[43] is an important question that has to be answered to fully elucidate the molecular mechanism of LE. A recent study proposed Brownian ratchet model for cohesin[44]. An analogy to the well-studied motor conventional kinesin-1 (Kin1), which walks towards the plus end of the stiff microtubule, is useful. It could be argued that Kin1 makes use of both power stroke and biased diffusion. Previous studies[45,46] showed that power stoke (neck-linker docking) propels the trailing head of Kin1 only by $\approx 5-6$ nm forward, which creates a strong bias to the next binding site. The rest of the step ($\approx 6-8$ nm) is completed by diffusion. Therefore, it is possible that both power stroke (sets the directionality) and Brownian ratchet are not mutually exclusive. Both the mechanisms could play a role in the LE process as well. Further structural, experimental, and computational studies are required to resolve fully the interplay between these mechanisms in LE.

**Conclusion**. We conclude with a few additional remarks. (1) We focused only on one-sided loop extrusion (asymmetric process) scenario for a single condensin, as demonstrated in the in vitro experiment[9]. Whether symmetric LE could occur when more than one condensin loads onto DNA producing Z-loop structures[14], and if the LE mechanism depends on the species[47] is yet to be settled. Similar issues likely exist in loop extrusion mediated by cohesins[48,49]. We believe that our work, which only relies on the polymer characteristics of DNA and on an allosteric (action at a distance) mechanism for loop extrusion, provides a framework for theoretical investigation of LE, accounting for different scenarios. (2) The $f$ dependence of LE velocity allows us to estimate the time scale for compacting the whole genome. In particular, if the loop extrusion velocity at $f = 0$ is taken to be ~1 kbp/s, we can calculate the LE time using the following assumptions. The number of condensin I and condensin II that are likely bound to DNA is ~3000 and ~500, respectively[50]. Therefore, the loops in the entire chromosome1 (~250 Mbps) could be extruded in a few minutes with the motors operating independently. The assumption that the motors operate independently is reasonable because the linear density (number of motors per genomic base pair) of the bound motors is low. A similar estimate has been made for loop extrusion time by cohesin in the G1 phase of HeLa cells[49]. These times are faster than the time needed to complete mitosis (~an hour)[51]. (3) Finally, if LE occurs by scrunching, as

gleaned from simulations, and advocated through experimental studies[19], it would imply that the location of the motor is relatively fixed on the DNA and the loop is extruded by ATP-driven shape transitions in the coiled-coils.

## Methods

**Analysis of experimental data.** For a given $L$, we sampled a Gaussian distribution for $R_1$ and generated 500 $R_1$ values. This procedure yields $P(L|R_1)$, allowing us to calculate the mean $R_1$. We repeated this procedure for different $L$ values. A similar procedure was use when $f \neq 0$. The step size distribution, $P(\Delta L)$, is $P(\Delta L) = P(L + R_2|R_1)$ where $P(L + R_2|R_1)$ is given in Eq. (1). Similarly, for $f \neq 0$, the step size distribution is $P(\Delta L|f) = P_A(L + R_2|R_1, f)$ where $P_A(L + R_2|R_1, f)$ is given in Eq. (3). The generated discrete distributions are smoothed by box moving average. The Mathematica notebook containing the demonstration of the main results is available, see "Code availability" section. The notebook contains: LE velocity ($\Omega$), $\Omega$ as a function of ATP concentration, $P(L|R)$, $P(L|R, f)$, $P(\Delta L)$, and $P(\Delta L|f)$.

Experimental data are digitized using Webplotdigitizer[52]. All the analysis for the experimental LE data is conducted using Wolfram Mathematica 12[53].

**MD simulations.** In order to study the O–B transition, we performed coarse-grained MD simulations as detailed in the supplementary information. The integration timestep was set to $\Delta t_L = 0.01 \tau_L$, where $\tau_L = 0.4 \sqrt{m\sigma^2/k_B T}$. For simplicity, we set mass $m = 1$, $\sigma = 1$ and $k_B T = 1$. We simulated 20,000 steps with Condensin in either B or O conformation. We repeated these simulations 50 times to ensure that the calculations have converged. The simulations were performed using OpenMM ver7[54]. The initialization script was written in-house in python 3.7 and has been uploaded to Github (see "Code availability" section). The Hamiltonian used for the simulation and parameters are given in the SI. We used MDTraj 1.9.5[55] for the analysis of the trajectories. We visualized the trajectories using VMD 1.9.5[56]. For sequence and structural alignment we used MultiSeq[57].

**Reporting summary.** Further information on research design is available in the Nature Research Reporting Summary linked to this article.

## Data availability

The data that support this study are available from the corresponding author upon reasonable request. The trajectory data from MD simulation generated have been deposited at https://bit.ly/3yfFRKg. The digitized experimental data[9,20,58] are available at https://github.com/kibidanngo/Condensin-Analysis_Experiments. In the SI, we used crystal structure for condensin, 6YVU and 6YVD, for normal mode analysis. The PDB structures are provided with this paper[40]. Source data are provided with this paper.

## Code availability

Molecular dynamics code for condensin simulations is deposited to Github: https://github.com/biofizzatreya/Condensin. Mathematica notebook for our theoretical results to analyze LE experiments is available https://github.com/kibidanngo/Condensin-Analysis_Experiments.

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

## Acknowledgements
We thank Rasika Harshey, Changbong Hyeon, Mauro Mugnai, and Johannes Stigler for useful comments and discussions. We are grateful to John Marko for clarifying some aspects of his model. This work was supported by NSF (CHE 19-00093), NIH (GM - 107703) and the Welch Foundation Grant F-0019 through the Collie-Welch chair.

## Author contributions
R.T. and D.T. conceived the theory, R.T., A.D., G.S. and D.T. performed the calculations and simulations, the experiment(s), R.T., A.D., G. S. and D.T. analyzed the results. All authors reviewed the manuscript.

## Competing interests
The authors declare no competing interests.
