## [Peer Review File · Nature Communications]

REVIEWER COMMENTS

Reviewer #1 (Remarks to the Author):

In this manuscript, the authors introduced an analytically solvable model for loop extrusion by condensin. The model describes one-side loop extrusion caused by a protein conformational change driven by ATP hydrolysis. Theoretical expressions of the model are simple and elegant and depend only on two parameters. The expressions fit well with experimental measurements of the dependence of loop extrusion velocity on external loading force. The value for condensin conformational change obtained from fitting agrees well with that from AFM experiments. The model also predicts the distribution of loop extrusion length that matches nicely with experimental measurements. Overall, I believe this paper presents a significant advancement in understanding the molecular mechanisms of loop extrusion. While I support its publication, I do have several comments that could help clarify the presentation.

First, I feel that more biological/molecular context could be provided for the model presented in Figure 1. Is the model first proposed by the authors, or have similar models already been introduced in the literature? Should I interpret A as binding with two strands of DNA as illustrated on the right? How might that occur at a molecular level? Why would A remain stationary as B explores and binds DNA? Is it because the DNA binding energy is somehow much stronger?

Are the authors envisioning a uni-directional extrusion? For example, would the loop size always increase over time? This appears to be the case, as dL in Figure 4d is always positive. It is not clear to me how that could arise if B randomly diffuses and binds with DNA.

It might be helpful to use a consistent coloring scheme for the structural model and the diagram on the right. For example, the authors color A in blue, which should correspond to the heads. But the hinge region is colored in blue on the left.

A reference to the estimation of the SMC less than 50 nm would be useful, potentially from some of the structural models.

For Figure 4d, I would probably avoid using red for $f=0$ pN to distinguish from the result shown in part c.

The authors used different values for the DNA persistence length when fitting with experimental data. More explanations are needed for their particular choices.

When estimating the distribution of loop extrusion length, the authors made significant assumptions regarding the length of the condensin in open and closed states. While I agree they are reasonable assumptions, there are also no direct evidence to support such assumptions. I would avoid very strong words as in the sentence “Remarkably, the calculated distribution is in excellent agreement with the experimental data”. Presumably, varying the assumptions on the distribution of R1 and R2 would impact the agreement between theory and experiment. If the theoretical prediction is independent of these assumptions, it is worth showing and emphasizing.

I am not sure I would agree with the following statement in the caption of Figure 5c “The agreement between the simulations and experiments is remarkable, especially considering that the model has no fitting parameters”. The persistence length of the coiled-coil domain is not in good agreement with the experimental value. In some sense, this persistence length is a parameter of the model/theory.

The variance for the O and B state appears comparable in the simulations, as shown in Figure 5b. This is in contrast to what was shown experimentally and the theory’s assumptions. The authors might want to clarify what dictates the variance in their simulations.

Reviewer #2 (Remarks to the Author):

Takaki and coworkers present a theoretical framework that predicts extrusion speeds and step lengths for condensin, explaining recent single molecule data by high-speed AFM and magnetic tweezers. Their theory is based on the transitions between within a two-state system of the condensin motor, driven by ATP, which alters the distance between two DNA-binding sites on the motor. Comparing their model to experimental data they predict that the size of the conformational change is ~22–26nm. Using course-grained simulations, the authors show that a possible mechanism for loop extrusion may be a scrunching mechanism, conveyed by conformational changes within the

ATPase heads that are propagated through the coiled-coil domains, suggesting that the two DNA binding sites may be the hinge and head domains.

The presented theory reproduces experimental data more accurately than previously published models, even though it relies on fewer parameters. It is therefore of great interest to the community for understanding processive loop extrusion by SMC complexes.

I am very much in favor of seeing this manuscript published, but there are some points that should be addressed:

1. It appears that the predicted mean step lengths deviate from experimental findings at high forces (Fig. S6). The authors state that the experimental data suffers from small sample sizes. However, there appears to be a systematic discrepancy that should be addressed.
2. Ref. 30 indicates that the persistence length of the coiled-coil arms is only $\sim 4\text{nm}$, while the suggested scrunching model appears to rely on stiff coiled coils (apart from the elbow). Are these notions compatible?
3. Is the suggested model equally applicable to SMCs other than condensin, especially to those whose coiled-coils do not appear to contain an elbow and seem to mostly adopt an I-shape (ref. 9)?
4. The proposed scrunching mechanism appears to be driven by the capture of DNA loops, where the captured loop size is mostly driven by thermal fluctuations. To understand this difference between SMCs and other motors which walk on stiff tracks, it would be nice to extend the corresponding discussion: Much of the literature on motor proteins talks about power strokes. Do SMC motors also exhibit a power stroke? Or are they rather Brownian ratchets that rely on thermal fluctuations of the substrate itself?
5. Is the size of the pre-existing loop L_0 relevant for the model? I assume that during initiation L_0 is small and the formation of L_0 may require bending energy.

Minor comments:

1. Ref. 22 is missing a title
2. Ref. 14 (a cohesin paper) is cited in the introduction in the context of condensin.

Reviewer #3 (Remarks to the Author):

Report on Takaki et. al. "The theory of condensin mediated loop extrusion in genomes"

This manuscript addresses key questions on how the ATP-dependent motor activity of the condensin SMC complex plays a role in DNA loop extrusion. Using simulations and analytical modelling, the authors explore the capabilities of the 'scrunching model' where condensin extrudes a DNA loop using a cyclic transition between open and closed states. They obtain quantitative results for the loop extrusion velocity, step size distributions, and hinge-head distances, which are all in remarkably good agreement with recent single-molecule experimental results, which is impressive and all the more so because this is obtained with only 2 or even zero fit parameters.

I find the work interesting and timely. There are hardly any other modelling studies around (currently mostly the work by Marko et al – Ref. 16 and since a week also a new preprint on BioRxiv: <https://www.biorxiv.org/content/10.1101/2021.03.15.435506v1>), while there is a great need in the field for modelling the mechanistic of DNA loop extrusion by SMC proteins which is one of the fundamental drivers of chromosome organization. I would support publication of the work if the authors can satisfactorily address the points below.

Major comments:

1. On page 1 the authors write 'we discovered a scrunching mechanism'. This is misleading since it suggests that the authors discovered this as a new mechanism. However, a scrunching model was already proposed by Kapanides et al for RNA polymerase (Ref.17), and proposed for SMC-driven DNA loop extrusion by Ryu et al (Ref.29). This should be properly acknowledged.

2. In the SI, the authors comment on the persistence length of the SMC arms in the condensin holocomplex (Ref.29). from their fits they estimate a value of 24 nm. This is a factor 6.3 larger than the experimental estimate of 3.8 nm (Ref.30) (not 5-fold larger as mentioned). The authors should discuss this discrepancy. In principle, for the case of the holocomplex, the dimerized head domains or Brn1 can constrain the SMC arms and hence the persistence length for the holocomplex images can indeed have a larger value. Yet, a factor of 6 is a much larger difference than expected. The authors should discuss in the main text, and not hide this in SI. Furthermore it is of interest to see how the fits like Fig.S3 would look for a cc persistence length of 4 nm (likely very poor).

3. The authors assumed that the allosteric transitions by ATP binding are effectuated through the movement of the elbow region. I would like to hear from the authors what role the thermal fluctuations do play in their model.

4. Following up on this point, it would be nice if the authors would generalize the model. For example, the authors should also explore the alternative scenario where no elbow exists and the flexible SMC arms exhibit a random diffusional search of the hinge for grabbing novel DNA. The authors should simulate this scenario, and discuss which scenario is more reasonable.

5. For the LE velocity (Fig. 4 and Fig. S5), a length conversion value of 0.34 nm per base pair was used, which only applies at very large forces (tens of pN), but essentially not in the force range used for the simulation. How would the prediction change with respect to the experimental data if force-dependent base pair length differences would be included in the simulation? More specifically, how will this affect the model prediction and experimentally determined genomic DNA length that is extruded at each LE step?

Minor comments:

1. Marko et al published a new preprint on BioRxiv last week:

<https://www.biorxiv.org/content/10.1101/2021.03.15.435506v1>),

and it would be good if the authors can comment on this in the paper.

2. An alternative scrunching-like mechanism for LE has recently been considered by Higashi et al, see www.biorxiv.org/content/10.1101/2021.02.14.431132v1. This alternative model is of interest simulate and compare. This might potentially determine which of the two models best resembles the experimental data.

3. I find the title too general. I would suggest to specify that the authors explore the scrunching model.

4. Why do (in Fig. S5) lower DNA persistence lengths lead to lower LE rates? At lower persistence lengths, one would expect that longer genomic DNA lengths would be extruded at low forces.

5. What is the reason for the prediction showing a non-intuitively smaller DNA step size peak at $f=0$ than predicted for higher DNA stretching forces in Fig. 4d?

6. It would be useful to reference more clearly to the individual Figure panels, and especially adding clear references for each SI figure.

7. The authors should critically consider their use of the wording of 'contour length'. In some cases, the use of 'DNA end-to-end length' would be a more correct description.

8. In Fig. 4d, it may be useful to add the DNA length distribution for 0.4 pN, as used for Fig. 4c to see potential differences more distinctively.

9. At the point where the authors compare the model prediction with experimentally determined LE step sizes, they might also include a related publication (<https://doi.org/10.1016/j.cub.2004.04.038>). While that publication only measured the step size at a fixed force of 0.4 pN, the step size distribution also follows a similar trend to the mentioned publication from Ryu et al., 2020.

10. There is a typo on page 11.: cohesion \square cohesin.

11. The layout of the figures should be improved (e.g. unreadable small fonts in Fig.5a-right).

12. Shouldn't the formula in the top line on page 5 have a minus sign in the exponent?

13. It would be good to refer in vivo observations that further support the scrunching model (Xiang and Koshland, 2021). In addition, Cryo-EM studies that support the conformations (because they also suggest the hinge engagement) should be acknowledged: Collier et al., 2020; Higashi et al., 2020; Shi et al., 2020.

14. The first few lines of the paper can be written a bit more sharply (knots are irrelevant to mention; the family of SMC proteins can be introduced more precisely).

15. The section on 'translocation' on page could be dropped or very significantly shortened, as the authors of Ref.8 have convincingly argued that this translocation observed in Ref.7 was an artifact due to salt and buffer conditions.

Responses to Reviewer 1

Reviewer 1: In this manuscript, the authors introduced an analytically solvable model for loop extrusion by condensin. The model describes one-side loop extrusion caused by a protein conformational change driven by ATP hydrolysis. Theoretical expressions of the model are simple and elegant and depend only on two parameters. The expressions fit well with experimental measurements of the dependence of loop extrusion velocity on external loading force. The value for condensin conformational change obtained from fitting agrees well with that from AFM experiments. The model also predicts the distribution of loop extrusion length that matches nicely with experimental measurements. Overall, I believe this paper presents a significant advancement in understanding the molecular mechanisms of loop extrusion. While I support its publication, I do have several comments that could help clarify the presentation.

Response: We appreciate the positive comments of this Reviewer 1 and, of course, are very pleased that he/she recommends publication.

Reviewer 1: First, I feel that more biological/molecular context could be provided for the model presented in Figure 1.

Response: We appreciate this suggestion. In the revision, we include an explanation of our model that is set in biological/molecular context. We have added the following in the main text to link our model to structure, as suggested by the reviewer.

The structure of condensin-DNA complex in the LE active form is currently unavailable. However, cryo-EM structures for the related cohesin-DNA complex [1] reveal that DNA is tightly gripped by the two heads of cohesin and the subunits (NIPBL and RAD21). When the results of structural studies are integrated with the observation that the hinge domain of the SMC complexes binds to DNA [2, 3, 4], we conclude that both condensin and cohesin must use a similar mechanism to engage with DNA. The head domains in these motors interact with the DNA segment that is in proximity whereas DNA binds only transiently to the hinge. We constructed the model in Fig.1 based in part on these findings.

Reviewer 1: Is the model first proposed by the authors, or have similar models already been introduced in the literature?

Response: We are the first to propose the model that could be translated into a viable theory for LE velocity and distribution of step lengths as a function of force. In the course of confirming some of the predictions of the theory using simulations, we found that LE does occur by the scrunching mechanism, which was first proposed in an unrelated context, the bubble formation in DNA mediated by bacterial RNA polymerase, the first step in transcription. In the context of loop extrusion a similar mechanism was proposed by Ryu et al. [5], which we show in our study naturally fits into our theory and simulations. Although we already related the scrunching in LE to transcription, we have added a sentence to make sure that the history of the discovery scrunching mechanism is correctly stated. Our work provides a firm theoretical basis.

In sharp contrast, by focusing on the motor activity of condensin through ATP-driven allosteric changes in the enzyme, our theory and simulations support "scrunching" as a plausible mechanism for loop extrusion. Scrunching is reminiscent of the proposal made over a decade ago in the context of the first stage in bacterial transcription that results in bubble formation in promoter DNA [6], which was quantitatively affirmed using molecular simulations [7]. Recently, the scrunching mechanism was proposed to explain loop extrusion [5], which is fully supported by theory and simulations presented here.

Reviewer 1: Should I interpret A as binding with two strands of DNA as illustrated on the right?

Response: The referee's interpretation is correct in that point A in Fig.1, the right panel, could be thought of two proximal DNA capturing/attaching sites of condensin.

Reviewer 1: How might that occur at a molecular level? Why would A remain stationary as B explores and binds DNA? Is it because the DNA binding energy is somehow much stronger?

Response: The referee is correct in that the DNA that is close to A, proximal to the two motor heads, is firmly grabbed by condensin. There is evidence that this is the case in the cryo-EM structure for the related cohesin [1] motor. This is in contrast to the contact at the hinge domain, which is likely due to transient binding to DNA.

Reviewer 1: Are the authors envisioning a uni-directional extrusion? For example, would the loop size always increase over time? This appears to be the case, as dL in Figure 4d is always positive. It is not clear to me how that could arise if B randomly diffuses and binds with DNA.

Response: We thank the referee for raising these important and subtle issues. Indeed, we are envisioning uni-directional extrusion. Mathematically the uni-directionality is incorporated into the theory in which there is asymmetry in the expressions for k^+ and k^- in Eq.(4), arising from the energy due to ATP hydrolysis μ . We believe that the negative loop extrusion length ($\Delta L < 0$) is unrelated to diffusion. If the negative loop extrusion is induced by the diffusional process, point B in Fig.1 would localize to the left of the point A in Fig. 1 (extruded DNA; L_0) of DNA. We think condensin avoids such a scenario by the orientation of the binding of smc2-4 and the subunits (Ycg1-Brn1) onto DNA. Although the possibility of decrease in loop size cannot be ruled out entirely, a structural study for cohesin [1] revealed that subunit (STAG1) interacts with the hinge domain of cohesin, which suggests that the direction of the binding of STAG1 may dictate the direction of the hinge movement. This in turn determines the directionality of the loop extrusion process. In other words, with high probability the capture of DNA by the hinge region occurs predominantly down stream of the motor heads, this ensuring that LE occurs predominantly in one direction. Of course, given that thermal fluctuations are important there is a finite probability, which we believe is small, of the motor slipping. We add the following discussion to describe directionality in the main text.

Directionality of LE: Directionality is a vital characteristic of biological molecular machines with SMC proteins being no exception. In our model (Fig.1), the unidirectional LE arises during two stages. One is the DNA capture process itself and the other is the actual active extrusion process (State1 to State2 in Fig. 1). Directionality emerges as a result of anisotropic binding of the SMCs and the associated subunits to DNA. The structure of cohesin [1] suggests that the subunit (STAG1) interacts with the hinge of cohesin by interactions that are anisotropic. This implies that the directionality of LE could be set by the very act of binding of the condensin motor heads, which poises the hinge to preferentially interact with DNA downstream of the motor head, schematically shown as point A in Fig. 1. This in turn ensures that the probability of capture of DNA resulting in $\Delta L < 0$ is minimized.

In addition, in our theory there is an asymmetry in the expressions for k^+ and k^- in Eq.(4) arising from the free energy, μ , due to the ATP hydrolysis. This process leads to a decrease in the head-hinge distance, as seen in the liquid AFM images of condensin in action. However, we believe that, with small probability, $\Delta L < 0$ could arise because of the slippage of DNA from the extruded DNA by the strong resistive force on the motor [8]. This situation is reminiscent of the slippage of

kinesin on microtubule under resistive load [9]. In the absence of any opposing force, the probability of $\Delta L < 0$ is likely to be small.

Reviewer 1: It might be helpful to use a consistent coloring scheme for the structural model and the diagram on the right. For example, the authors color A in blue, which should correspond to the heads. But the hinge region is colored in blue on the left.

Response: This is a useful suggestion. We changed the color scheme in Fig.1.

Reviewer 1: A reference to the estimation of the SMC less than 50 nm would be useful, potentially from some of the structural models.

Response: We include the reference for the structure of condensin Ref.[10].

Reviewer 1: For Figure 4d, I would probably avoid using red for $f=0$ pN to distinguish from the result shown in part c.

Response: We use a different color for $f = 0$ pN, as recommended by the reviewer.

Reviewer 1: The authors used different values for the DNA persistence length when fitting with experimental data. More explanations are needed for their particular choices.

Response: We obtained the data for the distribution from Ref. [8]. These authors measured the persistence length of the DNA used in their experiment, which is 42 nm. Thus, we used $l_p^{DNA} = 42$ nm for the graphs reporting the distribution $P(\Delta L)$ to be consistent with their study. For all the other graphs where we do not know the persistence length, we use $l_p^{DNA} = 50$ nm since it is the canonical value for double stranded DNA. We have modified the text to clarify this point.

Distribution for the DNA extrusion length per step (ΔL) of condensin using $l_p^{DNA} = 42$ nm, which is the value reported in the experiment [8].

Figure 1: Red: $\Delta = 13$ nm (main text). Green: $\Delta = 9$ nm. Blue: $\Delta = 5$ nm. The histogram is from Ryu et al [8].

Reviewer 1: When estimating the distribution of loop extrusion length, the authors made significant assumptions regarding the length of the condensin in open and closed states. While I agree they are reasonable assumptions, there are also no direct evidence to support such assumptions. I would avoid very strong words as in the sentence “Remarkably, the calculated distribution is in excellent agreement with the experimental data”. Presumably, varying the assumptions on the distribution of R_1 and R_2 would impact the agreement between theory and experiment. If the theoretical prediction is independent of these assumptions, it is worth showing and emphasizing.

Response: The concern is valid and worth additional discussion. We made assumptions that are supported by our theory as well as experiments [5]. Indeed modifying the assumption could affect the shape of the distribution $P(\Delta L)$. We added the figures in SI (Fig.S6) to show the effect of the standard deviation for R_1 , namely the effect of Δ in $R_1 \pm \Delta$. We added here the distributions for different Δ in Fig.1.

Reviewer 1: I am not sure I would agree with the following statement in the caption of Figure 5c “The agreement between the simulations and experiments is remarkable, especially considering that the model has no fitting parameters”. The persistence length of the coiled-coil domain is not in good agreement with the experimental value. In some sense, this persistence length is a parameter of the model/theory.

Response: This is indeed correct. Since this issue was brought up by the other two referees as well it is worth commenting on this problem. This referee is correct that the persistence length is part of the theory and the simulations. The simulations are independent of the assumptions in the theory (no input from the theory is used in the simulations). Nevertheless, the predictions of the theory and simulations are consistent with each other, and the outcomes are in good agreement with experiments. What matters most, perhaps, direct experimental observables, which we reproduce well with a couple of parameters at most. Measurement of the persistence lengths of biomolecules, especially active motor systems, is not straightforward. There are a number of examples in the literature (for example polyproline and chromatin). DNA is an exception in that sense. Therefore, getting the coiled coil persistence length in the construct of the motor is by no means trivial.

Despite these (possibly not so satisfactory) reasons, we acknowledge that there indeed is a discrepancy between our estimate of the persistence length (24 nm) and the experimental estimate (~ 3.8 nm). This is the only calculated parameter that does not agree with the experimental estimate, an important issue that deserves further investigation. Although we were aware of this rather large difference the precise reasons are unclear at this point, which clearly requires additional studies. We explored different smaller values in the simulations, which compromised the agreement of key experimentally measured quantities. After we received the reports, we tried a different set of simulations (described in our response to the third Reviewer), which did not resolve the discrepancy.

Reviewer 1: The variance for the O and B state appears comparable in the simulations, as shown in Figure 5b. This is in contrast to what was shown experimentally and the theory’s assumptions. The authors might want to clarify what dictates the variance in their simulations.

Response: In our simulations, the variance in the distributions mainly stems from two factors. One is the flexibility of the elbow (ϵ_b^{El} in Table S1) and the other is the conformational constraint at the head (k_C in Table S1). We also would like to note that the residence times in the open and closed shapes also affect the variance/height of the distributions. This implies that unless we know the catalytic cycle of condensin and the associated structural changes, we may not be able to recover the measured distributions for $P(R)$ quantitatively. It is also worth pointing out that there is a paucity in the experimental data, despite the beautiful experiments by Dekker’s laboratory. As additional data become available, we think that the theory and simulations could be further refined. Nevertheless, it should be appreciated that other aspects of the distributions are reproduced **without adjusting a parameter**. This we believe is non-trivial.

Responses to Reviewer 2

Reviewer 2: Takaki and coworkers present a theoretical framework that predicts extrusion speeds and step lengths for condensin, explaining recent single molecule data by high-speed AFM and magnetic tweezers. Their theory is based on the transitions between within a two-state system of the condensin motor, driven by ATP, which alters the distance between two DNA-binding sites on the motor. Comparing their model to experimental data they predict that the size of the conformational change is $22\pm 26\text{nm}$. Using course-grained simulations, the authors show that a possible mechanism for loop extrusion may be a scrunching mechanism, conveyed by conformational changes within the ATPase heads that are propagated through the coiled-coil domains, suggesting that the two DNA binding sites may be the hinge and head domains.

The presented theory reproduces experimental data more accurately than previously published models, even though it relies on fewer parameters. It is therefore of great interest to the community for understanding processive loop extrusion by SMC complexes.

I am very much in favor of seeing this manuscript published, but there are some points that should be addressed:

Response: This Reviewer has succinctly summarized our findings. We greatly appreciate the positive recommendation of Reviewer 2. In this revision we tried our best to answer the concerns of this Reviewer.

Reviewer 2: 1. It appears that the predicted mean step lengths deviate from experimental findings at high forces (Fig. S6). The authors state that the experimental data suffers from small sample sizes. However, there appears to be a systematic discrepancy that should be addressed.

Response: The Reviewer correctly points out that the differences between theory and experiment is systematic at high forces on DNA. Especially the theoretically derived $P(\Delta L)$ overestimates the variance at high external loads. The variance in the theoretically derived expression for $P(\Delta L)$ derived stems from the standard deviation in R_1 , namely Δ for $R_1 \pm \Delta$. We show in the SI that the effect of changing Δ and illustrate that the variance of the $P(\Delta L)$ decreases as Δ decreases, which eases the systematic error by decreasing the variance of the distributions (High loads in Fig.S6). We could have obtained better agreement by varying Δ with external load but that would increase the number of parameters without providing any additional insights. We also felt, as stated in the paper, that the data at

higher forces were not sufficiently extensive for us to undertake the additional step of making the theory more cumbersome.

Reviewer 2:

2. Ref. 30 indicates that the persistence length of the coiled-coil arms is only 4nm, while the suggested scrunching model appears to rely on stiff coiled coils (apart from the elbow). Are these notions compatible?

Response: The discordance in the persistence length between theory and experiment for coiled coils, which did not escape our attention during submission and afterwards, was also raised by the two other reviewers. We had given it a great deal of thought and have performed additional calculations after receiving the reports as well. We have not figured out a satisfactory resolution to this issue, which clearly warrants further investigation as well as possibly new experiments. The details of our attempts to understand this issue are outlined in our response to the first and the third Reviewer.

Reviewer 2:

3. Is the suggested model equally applicable to SMCs other than condensin, especially to those whose coiled-coils do not appear to contain an elbow and seem to mostly adopt an I-shape (ref. 9)?

Response: To the best of our knowledge Bacterial SMC, condensins, and cohesins all have elbow domain in the coiled-coil, and also adopt the I shape. We think, as discussed by Dekker and coworkers, that the I shape is not the active state during loop extrusion, and are likely only observed in situations where the SMCs are not engaged with DNA, as is the case in Ref.[10]. Therefore, we believe that the elbow domain must play a significant role for all the SMCs. If this is proven to be correct, with further experiments in which condensin is caught in the act of extruding loops as was done by Dekker, our theory would be broadly applicable to other SMCs as well. (We note parenthetically that the bacterial SMC has not been observed in loop extrusion activity).

Reviewer 2:

4. The proposed scrunching mechanism appears to be driven by the capture of DNA loops, where the captured loop size is mostly driven by thermal fluctuations. To understand this difference between SMCs and other motors which walk on

stiff tracks, it would be nice to extend the corresponding discussion: Much of the literature on motor proteins talks about power strokes. Do SMC motors also exhibit a power stroke? Or are they rather Brownian ratchets that rely on thermal fluctuations of the substrate itself?

Response: We would like to thank review 2 for bringing up this important point. We believe that no matter what we write here in the form of an explanation the power stroke versus Brownian ratchet issue will not be easily settled, as it is in the better studied AAA+ motors and kinesin and dynein. As noted by this reviewer the captured DNA in our model is determined by the thermal fluctuations. To contrast the behavior between SMC motors with better studied but still not fully understood motors, let consider kinesin-1 (Kin1). Kin1 makes use of both "power stroke" and also biased diffusion to realize the unidirectional cargo transport. Although not completely settled, "power stroke" in Kin1 likely corresponds to neck-linker docking where the conformational change induced by ATP binding promotes neck linker to dock to the front head. This process creates the needed bias for the motor to move to the next binding site (5 – 6 nm advance of the trailing head) in the forward direction and the rest of the process (6 – 8 nm to the next binding site) is completed by diffusion. The track for Kin1 is microtubule, which is possibly the stiffest polymer in cells.

We believe that it is too early to decide if power stroke or Brownian ratchet or a combination (our preference) is the mechanism of condensin action. In condensin, the importance of Brownian ratchet cannot be ruled out the role of thermal fluctuations and the smaller persistence of DNA compared to actin or microtubule. At this juncture, we do not favor one mechanism over the other. It is likely that both the mechanisms could play a role in the extrusion process. Initial conformational transition is likely driven by "power stroke" poising the motor to capture DNA down stream. The rest could be controlled by Brownian ratchets. We add the discussion for Brownian ratchet and power-stroke, in addition to the uni-directional loop extrusion.

Directionality of LE: Directionality is a vital characteristic of biological molecular machines with SMC proteins being no exception. In our model (Fig.1), the unidirectional LE arises during two stages. One is the DNA capture process and the other is the actual active loop extrusion (State1 to State2 in Fig. 1). Directional loop extrusion emerges as a result of binding of the SMCs and the associated subunits to DNA via anisotropic interactions. Indeed, the structure of cohesin [1] suggests that the subunit (STAG1) interacts with the hinge of cohesin by interactions that are anisotropic. This implies that the directional LE could be set by the very act of binding of the condensin motor heads to DNA, which poises the hinge to preferentially interact with DNA downstream of the motor head, we show schematically as point A in Fig. 1. This in turn ensures that the probability of capture of DNA resulting in $\Delta L < 0$ is minimized.

In addition, in our theory there is an asymmetry in the expressions for k^+ and k^- in Eq.(4) arising from the free energy, μ , due to the ATP hydrolysis. This process leads to a decrease in the head-hinge distance, as seen in the liquid AFM images of condensin in action. However, we believe that, with small probability, $\Delta L < 0$ could arise because of the slippage of DNA from the extruded DNA by the strong resistive force on the motor [8]. This situation is reminiscent of the slippage of kinesin on microtubule under resistive load [9]. In the absence of any opposing force, the probability that $\Delta L < 0$ is likely to be small.

Brownian ratchet and power-stroke: Whether SMC proteins employ Brownian ratchet mechanism [11] or power stroke [12] is an important question that has to be answered to fully elucidate the molecular mechanism of LE. A recent study proposed Brownian ratchet model for cohesin [13]. An analogy to the well-studied motor conventional kinesin-1 (Kin1), which walks towards the plus end of the stiff microtubule, is useful. It could be argued that Kin1 makes use of both power stroke and biased diffusion. Previous studies [14, 15] showed that power stroke (neck-linker docking) propels the trailing head of Kin1 only by $\approx 5-6$ nm forward, which creates a strong bias to the next binding site. The rest of the step ($\approx 6-19$ nm) is completed by diffusion. Therefore, it is possible that both power stroke (sets the directionality) and Brownian ratchet are not mutually exclusive. Both the mechanisms could play a role in the LE process as well. Further structural, experimental and computational studies are required to resolve fully the interplay between these mechanisms in LE.

Reviewer 2:

5. Is the size of the pre-existing loop L_0 relevant for the model? I assume that during initiation L_0 is small and the formation of L_0 may require bending energy.

Response: The Reviewer is absolutely correct. Technically, L_0 does not play any role. It can be seen in Fig.3 in the main text, which shows the bending energy required for condensin as extrusion proceeds. We see the peak for W at the initial stage of the extrusion process ($L_0 = 0; L_\Sigma = L_1$). Although the effect of length L_0 is subsumed in k_0 , and does not appear explicitly in our theory, we believe W less than $\mu \approx 20k_B T$ guarantees the initiation of LE process for condensin by overcoming the bending energy.

Reviewer 2:

Minor comments: 1. Ref. 22 is missing a title 2. Ref. 14 (a cohesin paper) is cited in the introduction in the context of condensin.

Response: We very much appreciate these comments. We fixed these problems in the revised manuscript.

Responses to Reviewer 3

Reviewer 3:

This manuscript addresses key questions on how the ATP-dependent motor activity of the condensin SMC complex plays a role in DNA loop extrusion. Using simulations and analytical modelling, the authors explore the capabilities of the “scrunching model” where condensin extrudes a DNA loop using a cyclic transition between open and closed states. They obtain quantitative results for the loop extrusion velocity, step size distributions, and hinge-head distances, which are all in remarkably good agreement with recent single-molecule experimental results, which is impressive and all the more so because this is obtained with only 2 or even zero fit parameters.

I find the work interesting and timely. There are hardly any other modelling studies around (currently mostly the work by Marko et al Ref. 16 and since a week also a new preprint on BioRxiv: <https://www.biorxiv.org/content/10.1101/2021.03.15.435506v1>), while there is a great need in the field for modelling the mechanistic of DNA loop extrusion by SMC proteins which is one of the fundamental drivers of chromosome organization. I would support publication of the work if the authors can satisfactorily address the points below.

Response: We very much appreciate the positive statement of this Reviewer and importantly for the thoughtful reading of the manuscript and the comments. We have tried our best to address the important issues raised, and have done so successfully except for one problem related to the persistence length of the coiled coil of the motor (also addressed in response to the other Reviewers). The bottom line is that, despite considerable thought and trying various simulations, we are at present unable to resolve the discrepancy.

Reviewer 3:

1. On page 1 the authors write “we discovered a scrunching mechanism”. This is misleading since it suggests that the authors discovered this as a new mechanism. However, a scrunching model was already proposed by Kapanides et al for RNA polymerase (Ref.17), and proposed for SMC-driven DNA loop extrusion by Ryu et al (Ref.29). This should be properly acknowledged.

Response: We apologize for this erroneous statement. We were most certainly aware of the previous proposal by Kapanides. In fact, Chen, Darst, and Thirumalai had reported simulations supporting the scrunching mechanism for bacterial transcription about a decade ago. Also, we are aware that Ryu and company did propose this mechanism independently of Kapanides for LE. Clearly, this was an

error, which **should not have been made**. We modified the sentence in the following way.

In sharp contrast, by focusing on the motor activity of condensin through ATP-driven allosteric changes in the enzyme leading to LE, our theory and simulations support "scrunching" as a plausible mechanism for the loop extrusion. Scrunching mechanism is reminiscent of the proposal made over a decade ago in the context of bacterial transcription that results in bubble formation in promoter DNA [6], which was subsequently illustrated using molecular simulations [7]. Recently, the scrunching mechanism was proposed to explain loop extrusion [5], which is fully supported by theory and simulations presented here.

Reviewer 3:

2. In the SI, the authors comment on the persistence length of the SMC arms in the condensin holocomplex (Ref.29). from their fits they estimate a value of 24 nm. This is a factor 6.3 larger than the experimental estimate of 3.8 nm (Ref.30) (not 5-fold larger as mentioned). The authors should discuss this discrepancy. In principle, for the case of the holocomplex, the dimerized head domains or Brn1 can constrain the SMC arms and hence the persistence length for the holocomplex images can indeed have a larger value. Yet, a factor of 6 is a much larger difference than expected. The authors should discuss in the main text, and not hide this in SI. Furthermore it is of interest to see how the fits like Fig.S3 would look for a cc persistence length of 4 nm (likely very poor).

Response: We apologize that this aspect was not explicitly stated in the main text, as we had thought it was. First, this oversight has been fixed. Second, this is the only calculated parameter, which differs from experimental estimate. We thought a lot about the potential reasons even before submitting the article and even more so afterwards. After the reports were received, we attempted the following set of calculations. We used the distribution of the angle between the coiled coils (reported in Fig.3 of ref [16]) and obtained an angular potential. The resulting potential and our worm-like chain model was used in the simulations to calculate the persistence length. The values obtained were still higher than ~ 4 nm estimated in the experiments. It is unclear, at this juncture, for the discrepancy even though our study faithfully captures all the observable (LE velocity and step size distributions) in the experiment. It is possible, as this Reviewer suggests, that due to interaction between condensin and other proteins (Brn1 for example) could constrain the head movement, and thus stiffen the coiled coil. However, the discrepancy between theoretical predictions and simulations is too large to be explained by such effects. We are baffled by the persistence length difference. Resolving this issue is important, as it reflects the structural aspects of the motor

in executing LE, and clearly requires additional future investigation. A discussion is added in the main text.

Although our theory is in good with experiments for load-dependent LE velocity and distribution of step sizes, the calculated persistence length (24 nm) of the SMC coils is much higher than the value (~ 3.8 nm) estimated from analyses of AFM images in combination with simulations using the worm-like chain model [16]. Despite performing many simulations using a variety of polymer models, the origin of this discrepancy is unclear. If in the simulations the coiled-coil persistence is decreased, we are unable to reproduce the measurements of quantities that are directly monitored in single molecule experiments [5] (see Fig.S3). We believe that additional experiments and simulations based on higher resolution structures in different nucleotide states are required to close the gap.

Reviewer 3:

3. The authors assumed that the allosteric transitions by ATP binding are effectuated through the movement of the elbow region. I would like to hear from the authors what role the thermal fluctuations do play in their model.

Response:

This point is also related to the coiled coil persistence length. Even though the coiled coil persistence length is large thermal fluctuations are important. In addition, thermal fluctuations are extremely important in determining the step size distribution, which is related to interaction of the hinge with DNA. We already performed simulations (Panel (a) in Figure S3 with coiled coil $l_p^{CC} = 4\text{nm}$). The results show that agreement with experiments is poor. Therefore, we think this one particular issue needs to be reconciled. As stated before, this might required more of the structural details. Please also refer the section regarding Brownian ratchet added in the revised manuscript.

Brownian ratchet and power-stroke: Whether SMC proteins employ Brownian ratchet mechanism [11] or power stroke [12] is an important question that has to be answered to fully elucidate the molecular mechanism of LE. A recent study proposed Brownian ratchet model for cohesin [13]. An analogy to the well-studied motor conventional kinesin-1 (Kin1), which walks towards the plus end of the stiff microtubule, is useful. It could be argued that Kin1 makes use of both power stroke and biased diffusion. Previous studies [14, 15] showed that power stroke (neck-linker docking) propels the trailing head of Kin1 only by $\approx 5\text{-}6$ nm forward, which creates a strong bias to the next binding site. The rest of the step ($\approx 6\text{-}19$ nm) is completed by diffusion. Therefore, it is possible that both power stroke (sets the directionality) and Brownian ratchet are not mutually exclusive. Both

the mechanisms could play a role in the LE process as well. Further structural, experimental and computational studies are required to resolve fully the interplay between these mechanisms in LE.

Reviewer 3:

4. Following up on this point, it would be nice if the authors would generalize the model. For example, the authors should also explore the alternative scenario where no elbow exists and the flexible SMC arms exhibit a random diffusional search of the hinge for grabbing novel DNA. The authors should simulate this scenario, and discuss which scenario is more reasonable.

Response: We already performed these simulations, which are in Fig.S3 in the SI. As the effect of the elbow is removed, by making it flexible as the Reviewer suggests, we are unable to reproduce the experiments on the movements involved in the O \rightarrow B transition (the inset in the Panel (a) in Figure S3 with coiled coil l_p^{CC} =4nm). The results show that agreement with experiments is poor. Therefore, we used a kink in the elbow, with stiffer coiled-coil, inspired by the liquid AFM images, in order to facilitate the allosteric mechanism for LE.

Reviewer 3:

5. For the LE velocity (Fig. 4 and Fig. S5), a length conversion value of 0.34 nm per base pair was used, which only applies at very large forces (tens of pN), but essentially not in the force range used for the simulation. How would the prediction change with respect to the experimental data if force-dependent base pair length differences would be included in the simulation? More specifically, how will this affect the model prediction and experimentally determined genomic DNA length that is extruded at each LE step?

Response: We appreciate the reviewer’s concern on this issue. However, we do not fully able to comprehend the reviewer’s concern, maybe because of some misunderstanding on our part. For a fixed contour length, the size of 1 base pair should not depend on force. The misunderstanding may come from the notion of genomic length (contour length) and end-to-end distance of DNA. The contour length is given by $(N - 1)b$ where N is the total number of base pairs and the end-to-end distance is determined by the polymer properties and thermal fluctuations. We calculated b from the contour length. We provide a fuller explanation, which we hope suffices. The apparent discrepancy likely arises because in some experiments the end-to-end distance, which is force dependent, is used to estimate b rather the contour length. The use of f -dependent b would only affect the LE

velocity as a function of extension and not the LE velocity as a function of load. Because the latter is the quantity we calculated theoretically there would only be a qualitative effect on LE velocity as a function of f -dependent b , which this reviewer suggests.

Reviewer 3:

1. Marko et al published a new preprint on BioRxiv last week: <https://www.biorxiv.org/content/10.1101/2021.03.15.435506v1>), and it would be good if the authors can comment on this in the paper.

Response: Thank you for letting us know the paper, which was online only after our paper was online and submitted for review. We became aware of the paper only recently. Despite studying it, we are unable to understand the model exactly because the authors do not even report the energy function used in the simulations. Moreover, to the extent we understand it, the model is predicated on the I shape, which is ruled out in the experiments by the Dekker's group. As a result, we are unable to comment in any intelligent manner on this paper. We hope that the Reviewer will forgive us.

Reviewer 3:

2. An alternative scrunching-like mechanism for LE has recently been considered by Higashi et al, see www.biorxiv.org/content/10.1101/2021.02.14.431132v1. This alternative model is of interest simulate and compare. This might potentially determine which of the two models best resembles the experimental data.

Response: We became aware of this only after submitting our paper. This is an important simulation study from a structural perspective. We cite the paper appropriately in the revised version. We agree that it would be interesting to compare in detail the two models. But doing so would require considerable work, taking us beyond the current goal, whose focus is mainly on the theory supplemented by simulations. In the future, we are planning to undertake simulations by incorporating the available structures, following our previous studies on kinesin, dynein, and myosin.

Reviewer 3:

3. I find the title too general. I would suggest to specify that the authors explore the scrunching model.

Response: Because the main focus is on theory, rather than the detailed simulations that are certainly an important aspect of our work, we prefer to leave the title as is. We hope that this Reviewer will indulge us on this!

Reviewer 3:

4. Why do (in Fig. S5) lower DNA persistence lengths lead to lower LE rates? At lower persistence lengths, one would expect that longer genomic DNA lengths would be extruded at low forces.

Response: We appreciate the reviewer for raising this issue. It could appear to be a counter-intuitive. This is because we used the approximation, $R \approx L$, even when l_p changes. This approximation becomes inaccurate as l_p decreases as seen in Fig.S5a. In principle, the reviewer is correct in that when l_p decreases the captured length of DNA would increase but the rate also involves time. On the other hand, from Eq. (7) in the main text, we find at given x , the force acting on DNA increases with the decrease of l_p , which in turn, lowers Ω , as manifested in Fig.S5b. In practice, the value of Ω , at a given extension x , would be determined by an interplay between these factors.

Reviewer 3:

5. What is the reason for the prediction showing a non-intuitively smaller DNA step size peak at $f=0$ than predicted for higher DNA stretching forces in Fig. 4d?

Response: The smaller peak for $f = 0$ pN is a the manifestation of the heavy-tail distribution for $P(\Delta L)$. For $f = 0$ pN, the heavy-tail of the distribution is significant. The peak of the distribution becomes smaller compared to the cases for higher forces where we do not see such a heavy tail. This prediction would be difficult to test experimentally because of the constraint in pulling experiments force cannot be zero.

Reviewer 3:

6. It would be useful to reference more clearly to the individual Figure panels, and especially adding clear references for each SI figure.

Response: We agree and have done so in the revision.

Reviewer 3:

7. The authors should critically consider their use of the wording of "contour length". In some cases, the use of "DNA end-to-end length" would be a more correct description.

Response: As is the custom in polymer physics, we use contour length to mean $L = (N - 1)b$, as stated earlier. The end-to-end distance has to be calculated or better measured, and is a function of force. We believe that our use of the terminology is reasonable.

Reviewer 3:

8. In Fig. 4d, it may be useful to add the DNA length distribution for 0.4 pN, as used for Fig. 4c to see potential differences more distinctively.

Response: We picked those f values in Fig.4d based on the visibility for the graph. $P(\Delta L)$ for $f = 0.2$ pN overlaps to the one for $f = 0.4$ pN significant amount which decreases the visibility. The two curves are overlaid in the inset to Fig. 4d in the revised version.

Reviewer 3:

9. At the point where the authors compare the model prediction with experimentally determined LE step sizes, they might also include a related publication (<https://doi.org/10.1016/j.cub.2004.04.038>). While that publication only measured the step size at a fixed force of 0.4 pN, the step size distribution also follows a similar trend to the mentioned publication from Ryu et al., 2020.

Response: We thank the Reviewer for pointing this important paper out. Indeed, Hirano's experiments are very relevant, especially considering they were done long ago. We included their data to Fig.4c in the revised version.

Reviewer 3:

10. There is a typo on page 11.: cohesion i cohesin.

Response: We corrected it in the revised version.

Reviewer 3:

11. The layout of the figures should be improved (e.g. unreadable small fonts in Fig.5a-right).

Response: We appreciate the suggestion. We improve the visibilities for the figures in the revised version.

Reviewer 3:

12. Shouldn't the formula in the top line on page 5 have a minus sign in the exponent?

Response: We appreciate the reviewer's concern. The expression is correct without changing the sign. It may be seen from the fact that for higher f , the weight $e^{fR/k_B T}$ brings the value of $P(R, f, |L)$ to larger R .

Reviewer 3:

13. It would be good to refer in vivo observations that further support the scrunching model (Xiang and Koshland, 2021). In addition, Cryo-EM studies that support the conformations (because they also suggest the hinge engagement) should be acknowledged: Collier et al., 2020; Higashi et al., 2020; Shi et al., 2020.

Response: We would like to thank to the reviewer for letting us know the important references. We properly acknowledge these references in the revised manuscript.

Reviewer 3:

14. The first few lines of the paper can be written a bit more sharply (knots are irrelevant to mention; the family of SMC proteins can be introduced more precisely).

Response: This is an interesting suggestion. We have tried to compromise between what we want say and the Reviewer thinks we should say.

Reviewer 3:

15. The section on "translocation" on page could be dropped or very significantly

shortened, as the authors of Ref.8 have convincingly argued that this translocation observed in Ref.7 was an artifact due to salt and buffer conditions.

Response: We would like to thank to the reviewer for suggesting the argument against the translocation. We still believe that highlighting the translocation as an unlikely mechanism is worth mentioning, considering most of the computational (including the most recent Marko paper) and some experimental studies for the genome organization rely on the translocation mechanism. Following this Reviewer’s advice we shortened the description.

References

- [1] Zhubing Shi et al. “Cryo-EM structure of the human cohesin-NIPBL-DNA complex”. In: *Science* (2020).
- [2] Allen Chiu, Ekaterina Revenkova, and Rolf Jessberger. “DNA interaction and dimerization of eukaryotic SMC hinge domains”. In: *Journal of Biological Chemistry* 279.25 (2004), pp. 26233–26242.
- [3] Julia J Griese, Gregor Witte, and Karl-Peter Hopfner. “Structure and DNA binding activity of the mouse condensin hinge domain highlight common and diverse features of SMC proteins”. In: *Nucleic acids research* 38.10 (2010), pp. 3454–3465.
- [4] Aaron Alt et al. “Specialized interfaces of Smc5/6 control hinge stability and DNA association”. In: *Nature communications* 8.1 (2017), pp. 1–14.
- [5] Je-Kyung Ryu et al. “The condensin holocomplex cycles dynamically between open and collapsed states”. In: *Nature structural & molecular biology* (2020), pp. 1–8.
- [6] Achillefs N Kapanidis et al. “Initial transcription by RNA polymerase proceeds through a DNA-scrunching mechanism”. In: *Science* 314.5802 (2006), pp. 1144–1147.
- [7] Jie Chen, Seth A Darst, and D Thirumalai. “Promoter melting triggered by bacterial RNA polymerase occurs in three steps”. In: *Proceedings of the National Academy of Sciences* 107.28 (2010), pp. 12523–12528.
- [8] Je-Kyung Ryu et al. “Resolving the step size in condensin-driven DNA loop extrusion identifies ATP binding as the step-generating process”. In: *Available at SSRN 3728949* (2020).
- [9] Swathi Sudhakar et al. “Germanium nanospheres for ultraresolution picotensiometry of kinesin motors”. In: *Science* 371.6530 (2021).
- [10] Byung-Gil Lee et al. “Cryo-EM structures of holo condensin reveal a subunit flip-flop mechanism”. In: *Nature Structural & Molecular Biology* 27.8 (2020), pp. 743–751.

- [11] R Dean Astumian. “Thermodynamics and kinetics of a Brownian motor”. In: *science* 276.5314 (1997), pp. 917–922.
- [12] Charles V Sindelar et al. “Two conformations in the human kinesin power stroke defined by X-ray crystallography and EPR spectroscopy”. In: *Nature structural biology* 9.11 (2002), pp. 844–848.
- [13] Torahiko L Higashi et al. “A Brownian ratchet model for DNA loop extrusion by the cohesin complex”. In: *bioRxiv* (2021).
- [14] Zhechun Zhang, Yonathan Goldtzvik, and Dave Thirumalai. “Parsing the roles of neck-linker docking and tethered head diffusion in the stepping dynamics of kinesin”. In: *Proceedings of the National Academy of Sciences* 114.46 (2017), E9838–E9845.
- [15] Zhechun Zhang and D Thirumalai. “Dissecting the kinematics of the kinesin step”. In: *Structure* 20.4 (2012), pp. 628–640.
- [16] Jorine M Eeftens et al. “Condensin Smc2-Smc4 dimers are flexible and dynamic”. In: *Cell reports* 14.8 (2016), pp. 1813–1818.

REVIEWER COMMENTS

Reviewer #1 (Remarks to the Author):

The authors have addressed all of my comments and I now support its publication.

Reviewer #2 (Remarks to the Author):

The authors have answered all my questions/concerns and I recommend publication.

Regarding the authors' response to Reviewer 2, question 3

("To the best of our knowledge Bacterial SMC, condensins, and cohesins all have elbow domain in the coiled-coil, and also adopt the I shape."):

It appears that at least for some bacterial SMC complexes, such as Smc-ScpAB, there is no elbow [1].

[1] <https://doi.org/10.1016/j.celrep.2021.109051>

Reviewer #3 (Remarks to the Author):

Second report by Reviewer 3 on the revised manuscript NCOMMS-21-08827A by Thirumalai et al

Overall, I maintain my opinion that this is an interesting piece of work that potentially deserves to be published in Nature Comm, and as I said before: I would support publication of the work if the authors can satisfactorily address the points raised.

However, I was overall disappointed by the rebuttal and the revision, and somewhat surprised by the unwillingness of the authors to add further data or even their attempts to do so. Below, I list a number of examples that need further consideration.

In my opinion, the authors should make a much stronger attempt to seriously address these points. In the current state, I recommend against publication in Nature Comm.

4. Following up on this point, it would be nice if the authors would generalize the model. For example, the authors should also explore the alternative scenario where no elbow exists and the flexible SMC arms exhibit a random diffusional search of the hinge for grabbing novel DNA. The authors should simulate this scenario, and discuss which scenario is more reasonable.

Response: We already performed these simulations, which are in Fig.S3 in the SI. As the effect of the elbow is removed, by making it flexible as the Reviewer suggests, we are unable to reproduce the experiments on the movements involved in the O-B transition (the inset in the Panel (a) in Figure S3 with coiled coil $l_{pcc} = 4\text{nm}$). The results show that agreement with experiments is poor. Therefore, we used a kink in the elbow, with stiffer coiled-coil, inspired by the liquid AFM images, in order to facilitate the allosteric mechanism for LE.

It is unclear to me that the elbow incorporation was abandoned in the data of Fig.S3. If so, I suggest to note that with more clarity and emphasis.

5. For the LE velocity (Fig. 4 and Fig. S5), a length conversion value of 0.34 nm per base pair was used, which only applies at very large forces (tens of pN), but essentially not in the force range used for the simulation. How would the prediction change with respect to the experimental data if force-dependent base pair length differences would be included in the simulation? More specifically, how will this affect the model prediction and experimentally determined genomic DNA length that is extruded at each LE step?

Response: We appreciate the reviewer's concern on this issue. However, we do not fully able to comprehend the reviewer's concern, maybe because of some misunderstanding on our part. For a fixed contour length, the size of 1 base pair should not depend on force. The misunderstanding may come from the notion of genomic length (contour length) and end-to-end distance of DNA. The contour length is given by $(N - 1)b$ where N is the total number of base pairs and the end-to-end

distance is determined by the polymer properties and thermal fluctuations. We calculated b from the contour length. We provide a fuller explanation, which we hope suffices. The apparent discrepancy likely arises because in some experiments the end-to-end distance, which is force dependent, is used to estimate b rather than the contour length. The use of f -dependent b would only affect the LE velocity as a function of extension and not the LE velocity as a function of load. Because the latter is the quantity we calculated theoretically there would only be a qualitative effect on LE velocity as a function of f -dependent b , which this reviewer suggests.

I agree that the base-base distance does not change with applied force, and indeed this is not what I meant to say. Also, the contour length is fixed. But the end-to-end distance is not, which is the essential point. And this is not incorporated adequately in the work.

Specifically: The authors compare in e.g. Fig. 4a experimental data (red dots), measured in bp/s, with their simulation results, which was converted from nm to bp using 0.34 nm per bp (i.e. 34 nm = 100 bp). This comparison is simply wrong.

On a related note, I do not understand why the authors are saying that LE velocity as function of load is force-independent. It is not, as various experiments have shown.

I am not satisfied with the answer of the authors. This is a core point that needs to be corrected and explained correctly in the paper.

1. Marko et al published a new preprint on BioRxiv last week:

<https://www.biorxiv.org/content/10.1101/2021.03.15.435506v1>), and it would be good if the authors can comment on this in the paper.

Response: Thank you for letting us know the paper, which was online only after our paper was online and submitted for review. We became aware of the paper only recently. Despite studying it, we are unable to understand the model exactly because the authors do not even report the energy function used in the simulations. Moreover, to the extent we understand it, the model is predicated on the I shape, which is ruled out in the experiments by the Dekker's group. As a result, we are unable to comment in any intelligent manner on this paper. We hope that the Reviewer will forgive us.

I disagree. Just stating this without even the minimal efforts of discussing the model is unsatisfactory to me (and readers).

2. An alternative scrunching-like mechanism for LE has recently been considered by Higashi et al, see www.biorxiv.org/content/10.1101/2021.02.14.431132v1. This alternative model is of interest simulate and compare. This might potentially determine which of the two models best resembles the experimental data.

Response: We became aware of this only after submitting our paper. This is an important simulation study from a structural perspective. We cite the paper appropriately in the revised version. We agree that it would be interesting to compare in detail the two models. But doing so would require considerable work, taking us beyond the current goal, whose focus is mainly on the theory supplemented by simulations. In the future, we are planning to undertake simulations by incorporating the available structures, following our previous studies on kinesin, dynein, and myosin.

Again, like in the last point, I am disappointed by the authors' unwillingness to perform further simulations. For publishing this work in Nature Comm I would expect more of a motivation to at least figure out (by simulations) which of the currently debated models fit best to the experimental data.

3. I find the title too general. I would suggest to specify that the authors explore the scrunching model.

Response: Because the main focus is on theory, rather than the detailed simulations that are certainly an important aspect of our work, we prefer to leave the title as is. We hope that this Reviewer will indulge us on this!

I disagree. There are various problems with this title:

- It is too general
- 'theory' is too ambitious. The bulk of this paper is not a new theory but simulations.
- The word 'genomes' does not fit either. Better would be 'DNA'

4. Why do (in Fig. S5) lower DNA persistence lengths lead to lower LE rates? At lower persistence lengths, one would expect that longer genomic DNA lengths would be extruded at low forces.

Response: We appreciate the reviewer for raising this issue. It could appear to be a counter-intuitive. This is because we used the approximation, $R \approx L$, even when l_p changes. This approximation becomes inaccurate as l_p decreases as seen in Fig.S5a. In principle, the reviewer is correct in that when l_p decreases the captured length of DNA would increase but the rate also involves time. On the other hand, from Eq. (7) in the main text, we find at given x , the force acting on DNA increases with the decrease of l_p , which in turn, lowers Ω , as manifested in Fig.S5b. In practice, the value of Ω , at a given extension x , would be determined by an interplay between these factors.

The authors should consider that the force, and consequently the average force-dependent LE step rate, does not vary in the experiments that the authors referred to in their comparisons. They should discuss this discrepancy.

6. It would be useful to reference more clearly to the individual Figure panels, and especially adding clear references for each SI figure.

Response: We agree and have done so in the revision.

This is still insufficient. Except for the reference to Fig. S3, none of the other SI figures are referenced in the main text.

9. At the point where the authors compare the model prediction with experimentally determined LE step sizes, they might also include a related publication (<https://doi.org/10.1016/j.cub.2004.04.038>). While that publication only measured the step size at a fixed force of 0.4 pN, the step size distribution also follows a similar trend to the mentioned publication from Ryu et al., 2020.

Response: We thank the Reviewer for pointing this important paper out. Indeed, Hirano's experiments are very relevant, especially considering they were done long ago. We included their data to Fig.4c in the revised version.

I thank the authors for adopting this but the visualization of Hirano's data is poor.

10. There is a typo on page 11.: cohesion ĩ cohesin.

Response: We corrected it in the revised version.

Thanks. But the revised manuscript still has various typos here and there. I would suggest a more careful screening of the text.

11. The layout of the figures should be improved (e.g. unreadable small fonts in Fig.5a-right).

Response: We appreciate the suggestion. We improve the visibilities for the figures in the revised version.

Still the figures have very small fonts at some places (e.g. the insets are not readable), which certainly is nonoptimal.

Reviewer 1:

The authors have addressed all of my comments and I now support its publication.

New Response: We are pleased that this reviewer has recommended publication of our paper.

Reviewer 2:

The authors have answered all my questions/concerns and I recommend publication.

New Response: We are pleased that this reviewer has recommended publication of our paper.

Regarding the authors' response to Reviewer 2, question 3 ("To the best of our knowledge Bacterial SMC, condensins, and cohesins all have elbow domain in the coiled-coil, and also adopt the I shape."):

It appears that at least for some bacterial SMC complexes, such as Smc-ScpAB, there is no elbow [1].

[1] <https://doi.org/10.1016/j.celrep.2021.109051>.

New Response: We thank the reviewer for pointing this paper out.

Reviewer 3 :

Overall, I maintain my opinion that this is an interesting piece of work that potentially deserves to be published in Nature Comm, and as I said before: I would support publication of the work if the authors can satisfactorily address the points raised.

However, I was overall disappointed by the rebuttal and the revision, and somewhat surprised by the unwillingness of the authors to add further data or even their attempts to do so. Below, I list a number of examples that need further consideration.

In my opinion, the authors should make a much stronger attempt to seriously address these points. In the current state, I recommend against publication in Nature Comm.

New Response: We apologize that our previous responses were totally satisfactory. In the current version, we have tried further to clarify the manuscript, by addressing more precisely the concerns of the Reviewer.

The texts in red are from reviewer 3 and the blue color are our responses. The changes in the manuscript is in green.

4. Following up on this point, it would be nice if the authors would generalize the model. For example, the authors should also explore the alternative scenario where no elbow exists and the flexible SMC arms exhibit a random diffusional search of the hinge for grabbing novel DNA. The authors should simulate this scenario, and discuss which scenario is more reasonable.

Initial Response: We already performed these simulations, which are in Fig.S3 in the SI. Perhaps, we did not highlight these results clearly. If the effect of the elbow is removed, by making it flexible as the Reviewer seems to suggest, we cannot reproduce the experiments on the movements involved in the O-B transition (the inset in the Panel (a) in Figure S3). The results show that very poor agreement with experiments. Therefore, we used a kink in the elbow, with stiffer coiled-coil, inspired by the liquid AFM images, in order to facilitate the allosteric mechanism for LE.

It is unclear to me that the elbow incorporation was abandoned in the data of Fig.S3. If so, I suggest to note that with more clarity and emphasis.

New Response: In the inset of Fig.3a, we effectively eliminated the elbow by using the identical energy scale for the elbow and the coiled-coil in the simulation. In other words, no distinction is made between coiled-coil and the elbow. We confirmed that in this case, we do not find the scrunching mechanism. Thus, some flexibility in the elbow, with the stiffness in the rest of the coiled-coil, is essential for the scrunching motion of condensin. We added a the following discussion in the SI.

The results in the inset in Fig.S2(a) is calculated using $l_p^{CC} \sim 4$ nm. In this instance, the elbow effect is eliminated by setting $\epsilon_b^{CC} = \epsilon_b^{El} (4k_B T)$, making the entire CCs flexible. The simulations show that $\Delta R \approx 0$, which contradicts experiments. We infer that the existence of rigid portion of CC with flexibility in the elbow region is essential for the scrunching mechanism.

5. For the LE velocity (Fig. 4 and Fig. S5), a length conversion value of 0.34 nm per base pair was used, which only applies at very large forces (tens of pN), but essentially not in the force range used for the simulation. How would the prediction change with respect to the experimental data if force-dependent base pair length differences would be included in the simulation? More specifically, how will this affect the model prediction and experimentally determined genomic DNA length that is extruded at each LE step?

Initial Response: We appreciate the reviewer’s continued concern on this issue. However, we do not fully able to comprehend the reviewer’s concern, maybe because of some misunderstanding on our part. For a fixed contour length, the size of 1 base pair should not depend on force. The misunderstanding may come from us not being entirely clear in explaining the difference between genomic length (contour length = L_c) and the end-to-end distance of DNA. The contour length is given by $L_c = (N - 1)b$ where N is the total number of base pairs and the end-to-end distance is determined by the polymer properties (WLC), thermal fluctuations, and force. We calculated b directly the contour length.

The apparent discrepancy likely arises because in some experiments the end-to-end distance, which is force (f) dependent, is used to estimate b rather the contour length. The use of f -dependent b would only affect the LE velocity as a function of extension and not the LE velocity as a function of load. Because the latter is the quantity we calculated theoretically there would only be a qualitative effect on LE velocity as a function of f -dependent b , which this reviewer suggests.

I agree that the base-base distance does not change with applied force, and indeed this is not what I meant to say. Also, the contour length is fixed. But the end-to-end distance is not, which is the essential point. And this is not incorporated adequately in the work.

New Response: First, we agree with all the sentences in the preceding paragraph except for the last sentence. Second, we note that we *explicitly* accounted for the force-dependence of the end-to-end distance theoretically in order to calculate the LE velocity. The f -dependent LE velocity (see Eq. (6)) as a function of f is then translated into LE velocity as a function of the relative (to the contour length) end-to-end extension, x using Eq. 7. Our theory produces LE velocity as a function of f and the dependence of Ω on x is obtained using Eq. (7). Here, clearly the relative end-to-end distance, x , depends explicitly on f . In the Ganji et al. [1] experiment x is measured and the f dependence is obtained by relating x to f . Thus, the dependence of the LE velocity on variations in x is correctly treated in the theory.

We add the following sentences following Eq. (7) to explicitly point out the dependence of the end-to-end distance on f .

In order to calculate Ω as a function of the relative DNA extension, x , we use the expression [2, 3],

$$f = \frac{k_B T}{2l_p} \left[2x + \frac{1}{2} \left(\frac{1}{1-x} \right)^2 - \frac{1}{2} \right].$$

The dimensionless variable, x , is the f -dependent relative extension. In the Ganji et al. [1] experiment x is measured and the f -dependence of Ω is obtained by relating x to f .

Figure 1: Blue line is from our theory and the green dots were obtained by digitizing the experimental data (Figure 3J in [1]).

New Response: In the figure 1 we compare Ω as a function of x using our analytical expression to the digitized plot of Fig. 3J in Ganji et al. [1]. The results in the experimental paper used a numerical procedure that relates x to f . Despite the very different procedure used in our study and experiments, the results are in good agreement with each other, especially considering that digitization invariably produces some errors.

Specifically: The authors compare in e.g. Fig. 4a experimental data (red dots), measured in bp/s, with their simulation results, which was converted from nm to bp using 0.34 nm per bp (i.e. 34 nm = 100 bp) . This comparison is simply wrong.

New Response: We first note that all the results in Fig. 4 are **obtained using theory and not simulations**. We respectfully disagree with the Reviewer that our conversion is incorrect. The 0.34 nm was obtained using $b = .34 \text{ nm} = \frac{L_c}{(N-1)}$, where N is the number of base pairs. As the Reviewer correctly notes, both L_c and N are constant, and hence b is indeed a constant. If b is f -dependent then L_c , the contour length, would change with f , which clearly is not correct as the Reviewer points out as well.

On a related note, I do not understand why the authors are saying that LE velocity as function of load is force-independent. It is not, as various experiments have shown.

I am not satisfied with the answer of the authors. This is a core point that needs to be corrected and explained correctly in the paper.

New response: It is possible that the initial reply may have been confusing. We apologize if that is the case. Nowhere in the paper do we even suggest, let alone state, that the LE velocity is f -independent (see Fig. 4a). The whole point is we are able to calculate LE velocity as a function of f using theory.

1. Marko et al published a new preprint on BioRxiv last week: <https://www.biorxiv.org/content/10.1101/2021.03.15.435506v1>), and it would be good if the authors can comment on this in the paper.

Initial Response: Thank you for letting us know the paper, which was online only after our paper was online and submitted for review. We became aware of the paper only recently. Despite studying it, we are unable to understand the model exactly because the authors do not even report the energy function used in the simulations. Moreover, to the extent we understand it, the model is predicated on the I shape, which is ruled out in the experiments by the Dekker's group. As a result, we are unable to comment in any intelligent manner on this paper. We hope that the Reviewer will forgive us.

I disagree. Just stating this without even the minimal efforts of discussing the model is unsatisfactory to me (and readers).

New Response: We found the Marko paper in bioarxiv, which appeared long after our paper, hard to comprehend because it lacks substantial details of their model. For example, we could not find any energy function used in the simulation. Thus, we are not in a position to simulate the model, and we do not feel it is needed for our paper. What we can say that is that because simulations in the above cited paper are based on the pumping model we think their mechanism is is not plausible. Indeed, this conclusion was reached previously by Dekker's group as well. We mention the study in context of Marko et al. [4].

(3) In the picture underlying the DNA capture model [4] (referred to as the DNA pumping model elsewhere [5]), the distance between the head and the hinge changes very little, if at all. Such a scenario is explicitly ruled out in an experimental study by Ryu et. al., [5] in part because they seldom observe the I shape in the holocomplex by itself or in association with DNA. For this reason, we believe that the mechanism proposed in the recent simulation study [6] is unlikely to be viable.

2. An alternative scrunching-like mechanism for LE has recently been considered by Higashi et al, see www.biorxiv.org/content/10.1101/2021.02.14.431132v1. This alternative model is of interest simulate and compare. This might potentially determine which of the two models best resembles the experimental data.

Initial Response: We became aware of this also only after submitting our paper. This is an important simulation study from a structural perspective. We cite the paper appropriately in the revised version. We agree that it would be interesting to compare in detail the two models. But doing so would require considerable work, taking us beyond the current goal, whose focus is mainly on the theory supplemented by simulations. In the future, we are planning to undertake simulations by incorporating the available structures, following our previous studies on kinesin, dynein, and myosin.

Again, like in the last point, I am disappointed by the authors' unwillingness to perform further simulations. For publishing this work in Nature Comm I would expect more of a motivation to at least figure out (by simulations) which of the currently debated models fit best to the experimental data.

New response: The simulations reported (again after our paper was under review) has similarities to our approach. They employed simulations of a model with a large number of parameters to calculate LE velocity, which we did using theory with just two parameters. The value of ΔR (Eq. 6), which they used to fit the experimental data, appears to larger than experiments and our simulations. Just as in our study they adjusted the persistence length of the coiled coil to achieve agreement with experiments. We feel that simulating their model quantitatively, which they have already done, is not necessary.

3. I find the title too general. I would suggest to specify that the authors explore the scrunching model.

Initial Response: Because the main focus is on theory, rather than the detailed simulations that are certainly an important aspect of our work, we prefer to leave the title as is. We hope that this Reviewer will indulge us on this!

I disagree. There are various problems with this title: - It is too general - "theory" is too ambitious. The bulk of this paper is not a new theory but simulations. - The word "genomes" does not fit either. Better would be "DNA"

New Response: We respectfully disagree. Most of our work is **new theory for LE** (simulations used to describe scrunching is in Figure 5), and not simulations. The simulations were undertaken to support the theory and elucidate the scrunching mechanism proposed in this context by C. Dekker and company. In deference

to the Reviewer we changed the title to "Theory and Simulations of condensin mediated loop extrusion in DNA".

4. Why do (in Fig. S5) lower DNA persistence lengths lead to lower LE rates? At lower persistence lengths, one would expect that longer genomic DNA lengths would be extruded at low forces.

Initial Response: We appreciate the reviewer for raising this issue. It could appear to be a counter-intuitive. This is because we used the approximation, $R \approx L$, even when l_p changes. This approximation becomes inaccurate as l_p decreases as seen in Fig.S5a. In principle, the reviewer is correct in that when l_p decreases the captured length of DNA would increase but the rate also involves time. On the other hand, from Eq. (7) in the main text, we find at given x , the force acting on DNA increases with the decrease of l_p , which in turn, lowers Ω , as manifested in Fig.S5b. In practice, the value of Ω , at a given extension x , would be determined by an interplay between these factors.

The authors should consider that the force, and consequently the average force-dependent LE step rate, does not vary in the experiments that the authors referred to in their comparisons. They should discuss this discrepancy.

New Response: We are not certain what the reviewer 3 means by "average force-dependent LE step rate". In Fig.S5, no experiments are referred to.

We include in SI the following explanation for Fig.S5 to further clarify the point.

The decrease of Ω as l_p decreases can be deduced from the linear (small x) expansion of Eq.(7) in the main text. In this case, $f = \frac{3k_B T}{2l_p}$. Substituting this linear expansion in the expression for Ω [Eq.(6)] confirms that l_p decreases as Ω becomes smaller. Note that this hold for fixed extrusion length per step (26 nm).

6. It would be useful to reference more clearly to the individual Figure panels, and especially adding clear references for each SI figure.

Initial Response: We agree and have done so in the revision.

This is still insufficient. Except for the reference to Fig. S3, none of the other SI figures are referenced in the main text.

New Response: We added all of the the section numbers for the SI which are referred to. The reviewer 3 is correct that not all the SI figures are mentioned in the main text but specifying the section numbers suffices the purpose of the connectivity between the main text and the SI.

9. At the point where the authors compare the model prediction with experimentally determined LE step sizes, they might also include a related publication (<https://doi.org/10.1016/j.cub.2004.04.038>). While that publication only measured the step size at a fixed force of 0.4 pN, the step size distribution also follows a similar trend to the mentioned publication from Ryu et al., 2020.

Initial Response: We thank the Reviewer for pointing this important paper out. Indeed, Hirano’s experiments are very relevant, especially considering they were done long ago. We included their data to Fig.4c in the revised version.

I thank the authors for adopting this but the visualization of Hirano’s data is poor.

New Response: We added the inset for Hirano’s data for better visibility.

10. There is a typo on page 11.: cohesion \tilde{A}^- cohesin.

Initial Response: We corrected it in the revised version.

Thanks. But the revised manuscript still has various typos here and there. I would suggest a more careful screening of the text.

New Response: We tried our best to eliminate typos.

11. The layout of the figures should be improved (e.g. unreadable small fonts in Fig.5a-right).

Initial Response: We appreciate the suggestion. We improve the visibility for the figures in the revised version.

Still the figures have very small fonts at some places (e.g. the insets are not readable), which certainly is nonoptimal.

New Response: We increased the font size for the figures.

References

- [1] Mahipal Ganji et al. “Real-time imaging of DNA loop extrusion by condensin”. In: *Science* 360.6384 (2018), pp. 102–105.
- [2] John F Marko and Eric D Siggia. “Stretching DNA”. In: *Macromolecules* 28.26 (1995), pp. 8759–8770.

- [3] Michael Rubinstein, Ralph H Colby, et al. *Polymer physics*. Vol. 23. Oxford university press New York, 2003.
- [4] John F Marko et al. “DNA-segment-capture model for loop extrusion by structural maintenance of chromosome (SMC) protein complexes”. In: *Nucleic acids research* 47.13 (2019), pp. 6956–6972.
- [5] Je-Kyung Ryu et al. “The condensin holocomplex cycles dynamically between open and collapsed states”. In: *Nature structural & molecular biology* (2020), pp. 1–8.
- [6] Stefanos K Nomidis et al. “DNA tension-modulated translocation and loop extrusion by SMC complexes revealed by molecular dynamics simulations”. In: *bioRxiv* (2021).

REVIEWERS' COMMENTS

Reviewer #3 (Remarks to the Author):

I have thoroughly read the revised version (the third version of it) and I do agree that the clarity of the presentation has improved.

I can support publication of the work in Nature Comm